# A-kinase anchoring protein BIG3 coordinates oestrogen signalling in breast cancer cells

Tetsuro Yoshimaru[1], Masaya Ono[2], Yoshimi Bando[3], Yi-An Chen[4], Kenji Mizuguchi[4], Hiroshi Shima[5], Masato Komatsu[1], Issei Imoto[6], Keisuke Izumi[7], Junko Honda[8], Yasuo Miyoshi[9], Mitsunori Sasa[10] & Toyomasa Katagiri[1]

Approximately 70% of breast cancer cells express oestrogen receptor alpha (ERα). Previous studies have shown that the Brefeldin A-inhibited guanine nucleotide-exchange protein 3–prohibitin 2 (BIG3-PHB2) complex has a crucial role in these cells. However, it remains unclear how BIG3 regulates the suppressive activity of PHB2. Here we demonstrate that BIG3 functions as an A-kinase anchoring protein that binds protein kinase A (PKA) and the α isoform of the catalytic subunit of protein phosphatase 1 (PP1Cα), thereby dephosphorylating and inactivating PHB2. E2-induced PKA-mediated phosphorylation of BIG3-S305 and -S1208 serves to enhance PP1Cα activity, resulting in E2/ERα signalling activation via PHB2 inactivation due to PHB2-S39 dephosphorylation. Furthermore, an analysis of independent cohorts of ERα-positive breast cancers patients reveal that both BIG3 overexpression and PHB2-S39 dephosphorylation are strongly associated with poor prognosis. This is the first demonstration of the mechanism of E2/ERα signalling activation via the BIG3–PKA–PP1Cα tri-complex in breast cancer cells.

[1] Division of Genome Medicine, Institute for Genome Research, Tokushima University, 3-18-15 Kuramoto-cho, Tokushima 770-8503, Japan. [2] Division of Chemotherapy and Clinical Research, National Cancer Center Research Institute, 5-1-1 Tsukiji, Chuo-ku, Tokyo 104-0045, Japan. [3] Division of Pathology, Tokushima University Hospital, 3-18-15 Kuramoto-cho, Tokushima 770-8503, Japan. [4] National Institutes of Biomedical Innovation, Health and Nutrition, 7-6-8 Saito-Asagi, Ibaraki, Osaka 567-0085, Japan. [5] Division of Cancer Chemotherapy, Miyagi Cancer Center Research Institute, 47-1 Nodayama, Medeshimashiote, Natori, Miyagi 981-1293, Japan. [6] Department of Human Genetics, Institute of Biomedical Sciences, Tokushima University Graduate School, 3-18-15 Kuramoto-cho, Tokushima 770-8503, Japan. [7] Department of Molecular and Environmental Pathology, Graduate School of Medicine, Tokushima University Graduate School, 3-18-15 Kuramoto-cho, Tokushima 770-8503, Japan. [8] Department of Surgery, National Hospital Organization Higashitokushima Medical Center, 1-1 Ohmukai-kita, Ootera, Itano, Tokushima 779-0193, Japan. [9] Department of Surgery, Division of Breast and Endocrine Surgery, Hyogo College of Medicine, 1-1 Mukogawa-cho, Nishinomiya, Hyogo 663-8501, Japan. [10] Department of Surgery, Tokushima Breast Care Clinic, 4-7-7 Nakashimada-cho, Tokushima 770-0052, Japan. Correspondence and requests for materials should be addressed to T.K. (email: tkatagi@genome.tokushima-u.ac.jp).

Oestrogen (E2) has a crucial role in regulating the initiation, development and progression of breast cancer, with ~70% of all breast cancer cells expressing oestrogen receptor alpha (ERα)[1,2]. In these cells, the biological actions of E2 are mediated by both genomic effects on the transcriptional activation of nuclear ERα and non-genomic effects on the activation of signalling pathways via plasma membrane-associated ERα. In particular, genomic ERα activation is influenced by coactivators and corepressors that positively or negatively modulate ERα-mediated transcriptional activity. However, although the role of coactivators in E2-dependent ERα-positive breast carcinogenesis has been well established, the pathophysiological role of corepressors remains highly debated.

Prohibitin 2 (PHB2), also known as REA[3], functions as both a modulator of the E2/ERα signalling network and a corepressor of ERα; however, its abundant expression in ERα-positive breast cancers is not well understood. We previously reported that Brefeldin A-inhibited guanine nucleotide-exchange protein 3 (BIG3); Q5TH69 in UniProt KB annotation, which is exclusively overexpressed in the majority of breast cancers[4,5], interacts with PHB2 in the cytoplasm, thereby inhibiting E2-dependent translocation to the nucleus and plasma membrane, resulting in the constitutive activation of the E2/ERα signalling pathways[5–9]. However, the pathophysiological role of BIG3 in the inactivation of PHB2 suppressive activity in breast cancer cells has not been elucidated.

Accumulating evidence has revealed that other BIG family proteins (for example, BIG1 and BIG2) contain A-kinase anchoring protein (AKAP) sequences in their N-terminal regions that bind the regulatory subunits of protein kinase A (PKA) and the γ isoform of the catalytic subunit of protein phosphatase 1 (PP1Cγ). These findings suggest that BIG1 and BIG 2 contribute to the regulation of ADB ribosylation factor[10,11]. Anchoring proteins, such as AKAP, bind to the catalytic subunit of protein phosphatase 1 (PP1C) to regulate its activity[12]. Indeed, several multivalent anchoring proteins form a complex and simultaneously co-localize with serine/threonine protein phosphatases and protein kinases[12,13].

A sequence comparison of BIG family proteins revealed that BIG3 showed only 21% identity with BIG1 and BIG2 (ref. 14). However, a detailed in silico analysis predicted that, similar to BIG1 and BIG2, BIG3 contains several regions that bind to the RII subunit of cyclic AMP (cAMP)-dependent PKA. In addition, BIG3 has been reported to potentially interact with the α isoform of the catalytic subunit of protein phosphatase 1 (PP1Cα) in vitro[15], suggesting that BIG1, BIG2 and BIG3 function as AKAPs. Indeed, in addition to BIG1 and BIG2, BIG3 contains a canonical PP1Cα − binding motif 'RVxF' sequence[15]. On the other hand, in silico analysis showed that BIG3 has no other PP1Cα-binding motifs such as G/SILK or MyPhoNe; however, both BIG1 and BIG2 contain G/SILK motifs. Therefore, BIG3 has been annotated as PPP1R33 by the Hugo Gene Nomenclature (HGNC) and designated as a member of the phosphatase regulatory subunit family. However, the functional impact of BIG3 as a PPP1R33 remains unknown. Therefore, understanding the properties of BIG3 as an AKAP, including PPP1R33, is critical for further elucidating the E2-dependent cell proliferation of ERα-positive breast cancers.

Here we report the direct pathophysiological role of BIG3 as a novel AKAP that forms a complex with PKA to act as a catalytic subunit of PP1Cα in E2-dependent breast carcinogenesis. The enhancement of PKA activity by E2 stimulation leads to the phosphorylation of BIG3 and suppresses the inhibition of PP1Cα activity, resulting in the dephosphorylation and inactivation of PHB2, which is essential for tumour suppressor activity. These findings strongly suggest that BIG3 is a critical coordinator of E2 signalling, resulting in an apparent 'loss of function' of the PHB2 protein in ERα-positive breast cancer cells.

## Results

**BIG3 forms a complex with PP1Cα and PKA.** In previous studies examining the detailed biological functions of BIG3 in the E2-signalling of ERα-positive breast cancer cells[5–9], we focused on the role of BIG3 as an in vitro PP1Cα interactor that contains a canonical PP1Cα–binding motif 'RVxF' sequence (1,228-KAVSF-1,232) (ref. 15), but not other PP1C-binding motifs such as G/SILK or MyPhoNe. We detected an endogenous interaction between BIG3 and PP1Cα in the ERα-positive breast cancer cell lines MCF-7 and KPL-3C, which highly express both proteins (Supplementary Fig. 1a), regardless of the presence of E2 (Fig. 1a). We further confirmed that the FLAG-tagged BIG3 mutant, which lacks an RVxF motif (ΔPP1Cα), completely abolished the interaction with endogenous PP1Cα (Supplementary Fig. 1b), indicating an in vivo endogenous BIG3–PP1Cα interaction in breast cancer cells.

Here we investigated the effect of BIG3 on PP1Cα activity against p-nitrophenyl phosphate (pNPP) by transfecting wild-type (WT) and ΔPP1Cα-BIG3 constructs into HEK293T cells. This showed that the introduction of WT-BIG3 inhibited endogenous PP1Cα activity in a dose-dependent manner, while ΔPP1Cα-BIG3 did not (Supplementary Fig. 1c). Furthermore, BIG3 deletion by short interfering RNA removed these inhibitory effects in MCF-7 and KPL-3C cells, regardless of the presence of E2 (Supplementary Fig. 1d). Thus, BIG3 may function as a potent inhibitor of PP1Cα activity. However, we also observed that the co-immunoprecipitation of BIG3 and PP1Cα resulted in an E2-dependent increase in PP1Cα activity in both BIG3 and PP1α immunoprecipitates in siEGFP-transfected cells (Fig. 1b and Supplementary Fig. 1d), and that E2 stimulation significantly increased PP1Cα activity for at least 6 h in MCF-7 and KPL-3C cells (Supplementary Fig. 1e). Collectively, these findings suggest that the inhibitory effect of BIG3 against PP1Cα activity may be eliminated by E2 stimulation in breast cancer cells. Interestingly, PP1Cα expression was remarkably upregulated after E2 stimulation. In contrast, PP1Cα expression was inhibited by treatment with tamoxifen, a selective ERα modulator at both the protein and transcriptional levels (Supplementary Fig. 1f). To obtain direct evidence for the upregulation of PPP1CA expression by E2/ERα, we performed a ChIP assay with E2-stimulated MCF-7 cells. The results showed that E2-dependent ERα recruitment was associated with the 5'-oestrogen response element (ERE) of the PPP1CA gene in MCF-7 cells (Supplementary Fig. 1f). As expected, E2 stimulation resulted in robust luciferase activity in cells transfected with the construct containing 5'-ERE from PPP1CA (Supplementary Fig. 1f), suggesting that PP1Cα activation is very long-lasting because it is a potential ERα-target gene.

To elucidate the mechanisms involved, we focused on the PKA-dependent phosphorylation of BIG3 as a key event occurring after E2 stimulation in breast cancer cells, as in silico analysis revealed that BIG3 contains two potential RII-binding domains (RIIBDs) that bind to the RII subunit of PKA (318–338 and 1,225–1,245 amino acids; Supplementary Fig. 1g), similar to BIG1 and BIG2 (refs 10,11), and PKA is highly expressed in all ERα-positive breast cancer cell lines (Supplementary Fig. 1h). We found that endogenous BIG3 interacts with PKA in breast cancer cells (Fig. 1c, Supplementary Fig. 1i) and that depletion of PKA led to almost complete suppression of the E2-induced serine and threonine phosphorylation levels of BIG3 (Fig. 1d). Notably, E2-induced PKA kinase activity was drastically enhanced compared with the E2-induced increased interaction of PKA

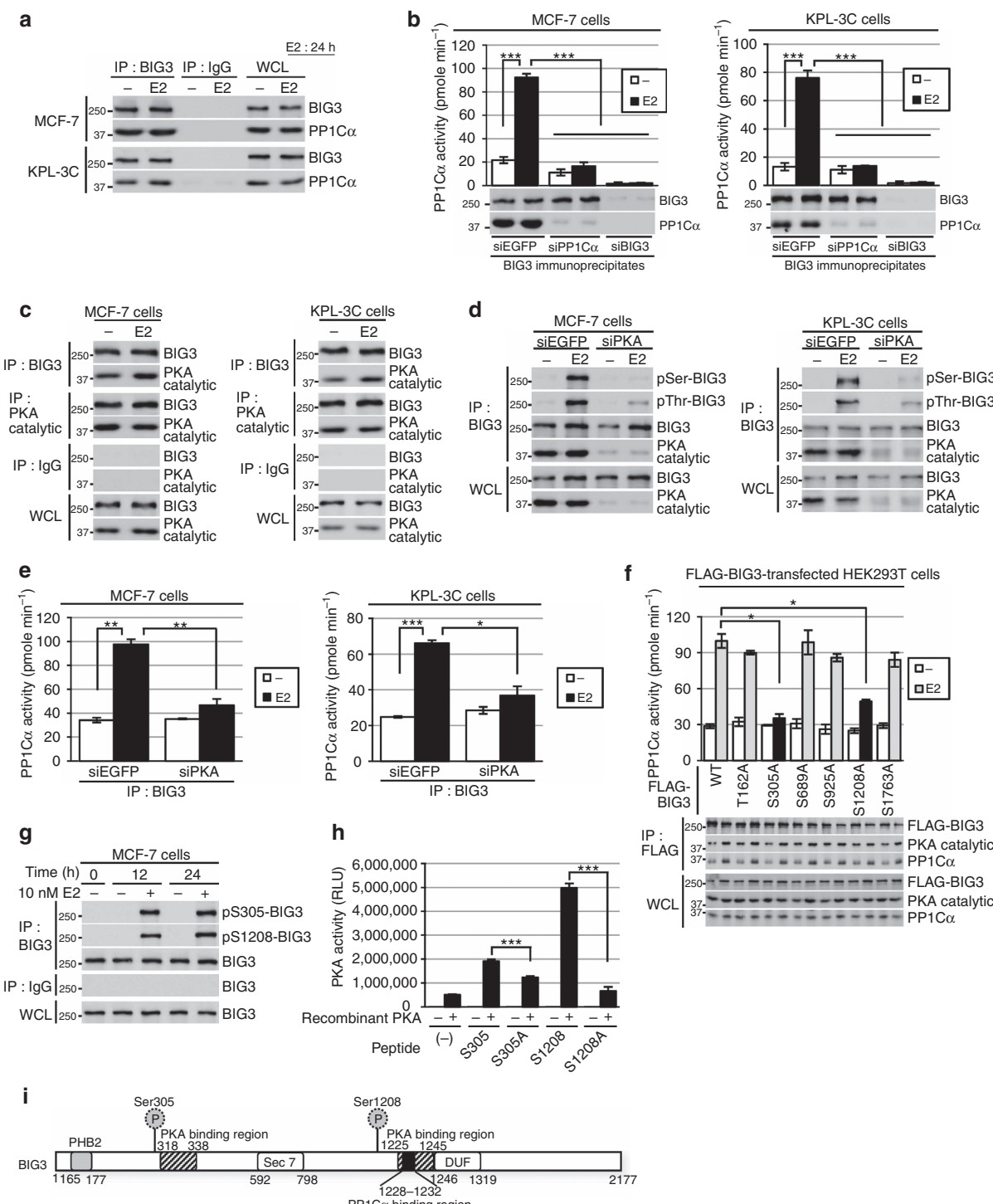

**Figure 1 | BIG3 phosphorylation functions as a regulatory subunit of PP1Cα.** (**a**) Interaction of endogenous PP1Cα with BIG3 in MCF-7 and KPL-3C cells after E2 stimulation for 24 h. (**b**) Effects of siPP1Cα and siBIG3 on phosphatase activity of BIG3 immunoprecipitates in MCF-7 and KPL3C cells. These data represent the means ± s.e.m. of three independent experiments. (**c**) Interaction of endogenous PKA catalytic subunit with BIG3 after E2 stimulation for 24 h in MCF-7 and KPL-3C cells. (**d**) The inhibitory effects of siPKA on BIG3 phosphorylation in MCF-7 and KPL-3C cells. Representative results are shown from one of three experiments. (**e**) The inhibitory effects of siPKA on PP1Cα phosphatase activity of BIG3 immunoprecipitates after E2 stimulation for 24 h in MCF-7and KPL-3C cells. These data represent the means ± s.e.m. of three independent experiments. (**f**) Analysis of PP1Cα phosphatase activity to identify the phosphorylation sites in BIG3 by PKA. The indicated FLAG-tagged BIG3 mutants (T162A, S305A, S689A, S925A, S1208A and S1763A) and HA-tagged ERα construct-transfected HEK293T cells were immunoprecipitated using an anti-FLAG antibody. The immunoprecipitates were immunoblotted, and the phosphatase activity against *p*NPP was measured. Each PP1Cα activity was normalized to the PP1Cα signal intensity of immunoprecipitates in FLAG-tagged BIG3 (WT)-transfected HEK293T cells. These data represent the means ± s.e.m. of three independent experiments. (**g**) BIG3 phosphorylation at S305 and S1208 at the indicated time points after E2 stimulation. (**h**) *In vitro* PKA activation assay of BIG3 using engineered peptides representing S305 and S1208. These data represent the means ± s.e.m. of three independent experiments. (**i**) The schematic representation of BIG3. PKA-binding regions: 318–338 and 1,225–1,245, PP1Cα − binding region: 1,228–1,232. *P < 0.05, **P < 0.01, ***P < 0.001 (two-sided Student's *t*-test).

with BIG3 (Supplementary Fig. 1i,j), suggesting that E2 stimulation led to the enhancement of PKA activity rather than the increase of its binding to BIG3. We next examined the knockdown effects of PKA on PP1Cα activity and observed that treatment with siPKA significantly abrogated E2-dependent

PP1Cα activity, while siEGFP did not (Fig. 1e), indicating that the PKA-dependent serine phosphorylation of BIG3 following E2 stimulation eliminates its inhibitory effect against PP1Cα activity.

To determine the PKA-mediated phosphorylation sites in BIG3, we focused on five serine and one threonine sites that were

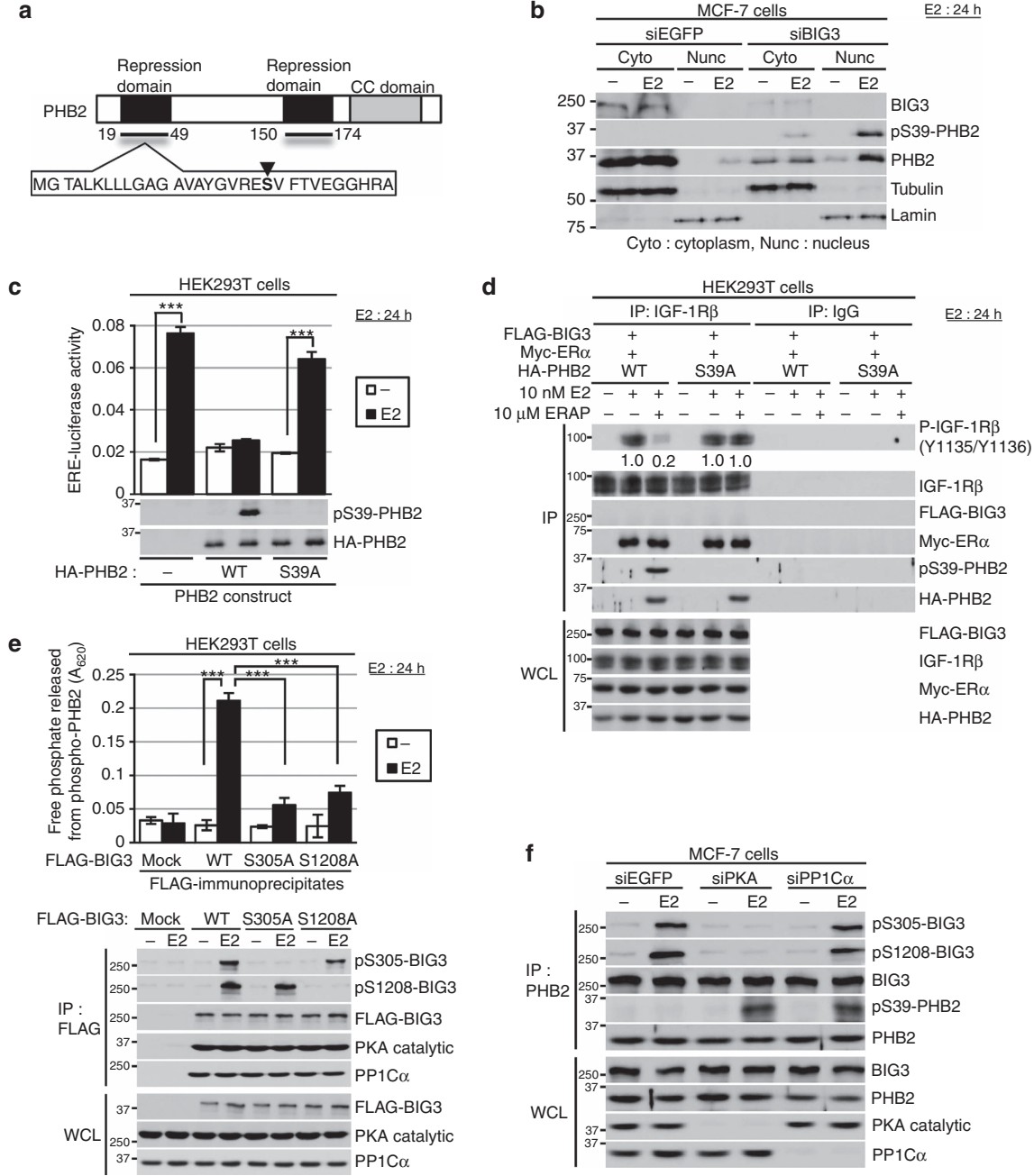

**Figure 2 | BIG3-PKA-PP1Cα regulates the suppressive ability of PHB2.** (**a**) The schematic representation of human PHB2. The black inverted triangle indicates S39 of PHB2. (**b**) The subcellular localization of phospho-PHB2 S39 after siBIG3 treatment in the presence of E2. α/β-Tubulin (tublin) and laminin B (lamin) were used as loading controls for the cytoplasmic and nuclear fractions, respectively. (**c**) Upper, Luciferase assays showing the inhibitory activity of phospho-PHB2 S39. The indicated PHB2 (WT, S39A), ERα, ERE-luciferase, and pRL-TK construct-transfected HEK293T cells were exposed to E2 for 24 h and analysed for luciferase and *Renilla*-luciferase activities. Data are normalized to *Renilla*-luciferase and represent the means ± s.e.m. of three independent experiments. Lower, PHB2 S39-phosphorylation. HA-PHB2 and FLAG-ERα-transfected HEK293T cells were stimulated by E2 for 24 h. (**d**) The inhibitory effects of PHB2-S39 on IGF-1Rβ phosphorylation. The indicated PHB2 (WT, S39A) or ERα-transfected HEK293T cells were exposed to E2 for 24 h, followed by immunoprecipitation with IGF-1Rβ antibody. (**e**) Phosphatase activities of BIG3 against phospho-PHB2-peptide. The indicated FLAG-BIG3 (WT, S305A and S1208A) and HA-ERα-transfected HEK293T cells were treated with E2 for 24 h, followed by immunoprecipitation using an anti-FLAG antibody. Phosphatase activity of the immunoprecipitates was measured as described in Methods section. These data represent the means ± s.e.m. of three independent experiments. (**f**) The effects of siPKA and siPP1Cα on BIG3 S305 and S1208 phosphorylation, and PHB2 S39-phosphorylation in MCF-7 cells after E2 stimulation for 24 h. ***P < 0.001 (two-sided Student's *t*-test).

predicted to have a higher potential using the NetPhos 3.1 server[16] ($\geq 0.78$; Supplementary Fig. 1k). We next examined the effects of these sites on PP1Cα phosphatase activity using HEK293T cells exogenously overexpressed with each FLAG-BIG3 mutant construct. The amounts of PP1Cα which bound to the exogenously expressed-FLAG-BIG3 in HEK293T cells was increased in the presence of E2 stimulation, suggesting the possibility that the E2-dependent PP1Cα phosphatase activation was depended on the increase of PP1Cα amounts, which binds to BIG3. On the other hand, among these sites, mutations in both S305 and S1208 (via substitutions with alanine), but not mutations in other sites significantly reduced E2-dependent PP1Cα activity against $p$NPP as a substrate regardless of the presence of high amount of PP1Cα. These findings suggest that both S305 and S1208 of BIG3 are important sites for PP1Cα phosphatase activity (Fig. 1f; Supplementary Fig. 1l). In contrast, pseudo-phosphorylation mutations at each serine residue (S305E and S1208E) and, in particular, double pseudo-phosphorylation mutations (S305E/S1208E) did not reduce PP1Cα activity (Supplementary Fig. 1l). Furthermore, the E2-induced phosphorylation level of BIG3-S305 and -S1208 was maintained for 24 h in MCF-7 cells (Fig. 1g), whereas it was almost completely suppressed by siPKA treatment (Supplementary Fig. 1m). These results suggest the importance of PKA-mediated phosphorylation of these particular sites in BIG3 for E2-induced PP1Cα activation. To confirm this, we performed an *in vitro* kinase assay and a two-dimensional image-converted analysis of liquid chromatography and mass spectrometry (2DICAL) with each synthesized peptide containing BIG3-S305 and -S1208. We found that the recombinant PKA directly phosphorylates the S305 and S1208 peptides, but not the S305A or S1208A peptides (Fig. 1h). We also detected the phosphorylation of both S305 and S1208 in BIG3 (DHGRG(pS)GCSCTAPALSGPVAR and RCW(pS) LVAPH; Supplementary Fig. 1n; Supplementary Table 1). This indicates that PKA directly phosphorylated BIG3 at these sites following E2 stimulation, removing its inhibitory effect against PP1Cα activity. Taken together, these findings strongly suggest that BIG3 forms a heterotrimeric signalling complex with PP1Cα and PKA, and functions as an AKAP in the presence of E2 in breast cancer cells (Fig. 1i).

**Phospho-PHB2-S39 is inhibitory**. We previously reported that the BIG3–PHB2 complex has a critical role in the E2/ERα signalling pathways in breast cancer cells[5–9]. Interestingly, here we observed that treatment with ERα activity-regulator synthetic peptide (ERAP), a dominant-negative peptide inhibitor that targets this interaction, led to the serine phosphorylation of PHB2 immunoprecipitated with nuclear ERα in the presence of E2 (Supplementary Fig. 2a), and also rapidly restored and maintained the E2-dependent serine phosphorylation of PHB2 for 24 h in MCF-7 cells (Supplementary Fig. 2b). Therefore, we hypothesized that interactions with the BIG3–PKA–PP1Cα tri-complex dephosphorylate the serine residues in PHB2.

To determine the candidate phosphorylation site in PHB2, we focused on S39 because it is located within the repression domain of ERα transcriptional activity (19–49 amino acids; Fig. 2a) and is potentially phosphorylated[3,17]. Western blot analysis using an S39 phosphorylation-specific antibody showed that E2-dependent S39 phosphorylation of nuclear and cytoplasmic PHB2 occurred in siBIG3-treated MCF-7 cells, but not in siEGFP-treated cells (Fig. 2b). To examine the effects of the phosphorylation status of PHB2 at S39 on ERα transcriptional activity, we co-transfected HEK293T cells with HA-tagged PHB2-WT or mutated PHB2, in which S39 was substituted with alanine (S39A), and FLAG-tagged ERα, and performed an ERE reporter assay. These results showed

that PHB2-WT, but not PHB2-S39A, released from BIG3 by ERAP treatment inhibit E2-induced ERα transcriptional activity (Fig. 2c). Immunocytochemical approaches using the anti-PHB2-specific antibody also revealed that phosphorylation of nuclear-translocated endogenous PHB2 at S39 occurred in ERAP-treated MCF-7 cells in the presence of E2 but completely disappeared following λ-phosphatase treatment (Supplementary Fig. 2c). These findings suggest that PHB2-S39 phosphorylation is required for the repression of E2-induced ERα transcriptional activity. On the other hand, we also demonstrated that the amount of PHB2-S39A protein that interacted with nuclear ERα was slightly reduced compared with the amount of PHB2-WT that interacted with nuclear ERα (Supplementary Fig. 2d). These results suggest that PHB2-S39 phosphorylation may also involve the E2-dependent nuclear translocation of PHB2 via possible additional or different phosphorylation events. Notably, PHB2-S39A abolished interference of the E2-induced tyrosine phosphorylation of IGF-1Rβ, and the interaction between IGF-1Rβ and ERα (Fig. 2d). Collectively, these findings suggest that the S39 phosphorylation of PHB2 is essential for its suppression of E2-induced genomic and non-genomic ERα signalling activation.

**BIG3-PP1Cα-PKA dephosphorylates PHB2-S39**. We examined the direct effect of PP1Cα activity in the BIG3-PP1Cα-PKA complex upon PHB2-S39 phosphorylation using an *in vitro* malachite green phosphatase assay (see Methods). We mixed immunoprecipitates of FLAG-BIG3 (WT)-transfected HEK293T cells with the phosphorylated PHB2-WT (S39) peptide as a substrate, and monitored the free phosphate that was released. E2 stimulation significantly increased the amount of free phosphate produced by BIG3-WT-transfected cells, but not BIG3-S305A and − S1208A transfected cells (Fig. 2e). BIG3-WT-S305 and S1208 were phosphorylated in the presence of E2, whereas BIG3-S305A and -S1208A were not, indicating that both S305 and S1208 phosphorylation are required to achieve maximal PHB2 phosphatase activity. Conversely, PKA depletion resulted in the complete inhibition of BIG3-S305 and -S1208 phosphorylation, and PP1Cα activity against PHB2-S39 phosphorylation in the presence of E2 (Fig. 2f; Supplementary Fig. 3a). PP1Cα depletion also recovered PHB2-S39 phosphorylation, but had no effect on BIG3-S305 and -S1208 phosphorylation in the presence of E2 (Fig. 2f; Supplementary Fig. 3b). We confirmed that the PKA–BIG3–PP1Cα complex inactivates PHB2 through the use of the BIG3–PP1Cα interaction inhibitor (Supplementary Fig. 3c), the PKA inhibitor H-89 and the PP1Cα inhibitor okadaic acid (Supplementary Fig. 3d). To investigate the direct effect of PP1Cα on PHB2-S39 phosphorylation, we performed *in vitro* phosphatase assays using a recombinant active PP1Cα and the phosphorylated PHB2-S39 peptide (YGVRE(pS)VFTVE) as a substrate, and conducted a mass spectrometry analysis. This confirmed that PP1Cα dephosphorylated the phosphorylated PHB2-S39 peptide (Supplementary Fig. 3e).

**PKCα phosphorylates PHB2-S39**. PKA and PKCα have previously been reported as potential kinases for PHB2 phosphorylation[3,18] and were abundantly expressed in ERα-positive breast cancer cell lines (Supplementary Figs 1h and 4a). Therefore, here we considered their role in PHB2-S39 phosphorylation. We found that short interfering RNA mediated the knockdown of PKCα, but not PKA, resulting in the complete inhibition of E2-induced PHB2-S39 phosphorylation in ERAP-treated MCF-7 and KPL-3C cells (Fig. 3a; Supplementary Fig. 4b). Subsequent *in vitro* kinase assays showed that PKCα phosphorylated the PHB2-WT peptide, but only minimally

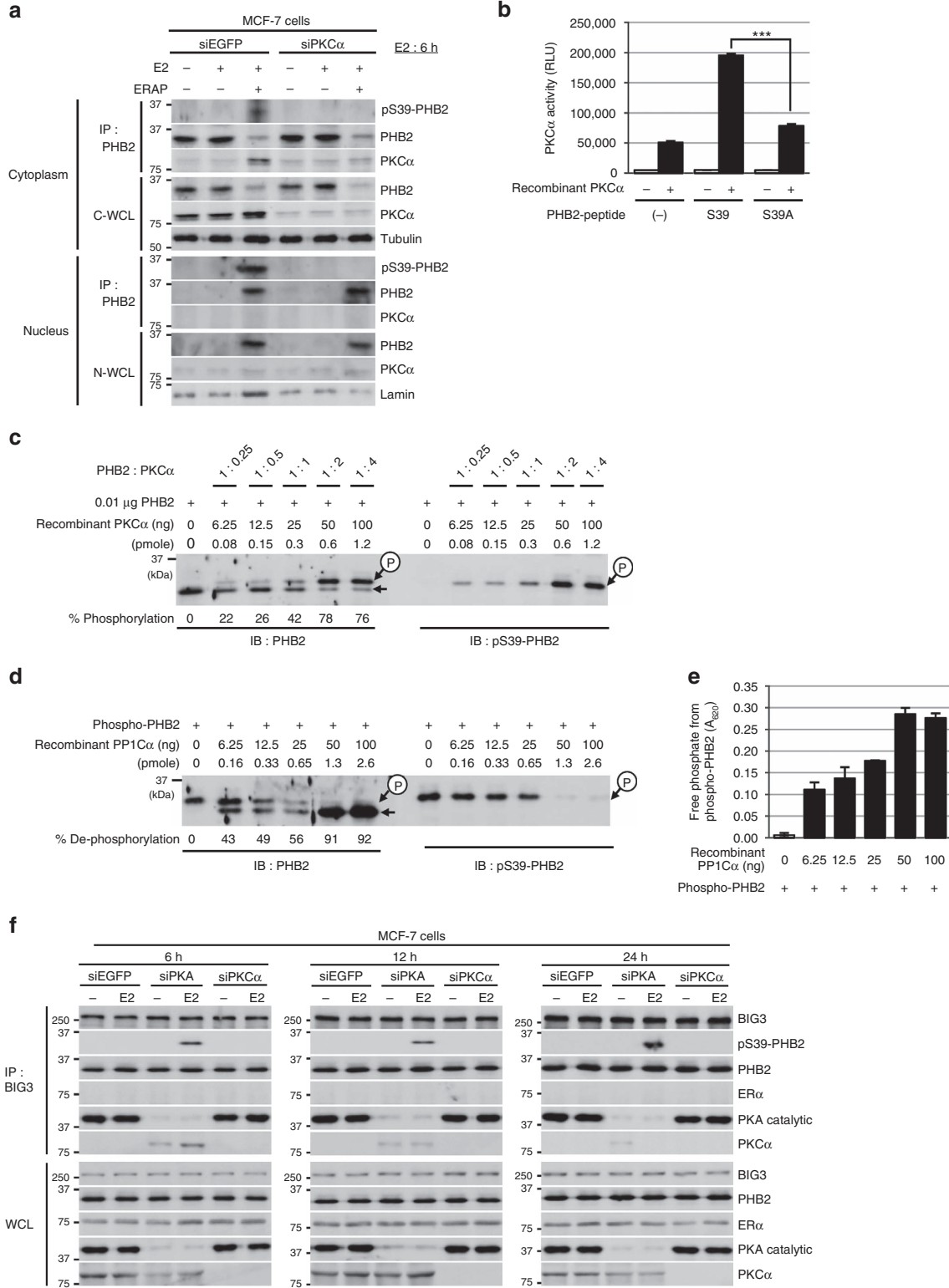

**Figure 3 | PHB2 is activated by phosphorylation at Ser39 via PKCα.** (**a**) The inhibitory effects of siPKCα on PHB2 phosphorylation at S39 in PHB2 immunoprecipitates of cytoplasmic and nuclear fractions in MCF-7 cells. C-WCL, total cytoplasmic lysates; N-WCL, total nuclear lysates. (**b**) *In vitro* PKCα activation assay of PHB2 using engineered peptides representing S39 and the alanine mutant of S39 (S39A). These data represent the means ± s.e.m. of three independent experiments. (**c**) Phosphorylation at S39 in the recombinant full-length PHB2 by PKCα using Phos-tag SDS–PAGE and western blot analysis with anti-PHB2 and anti-pS39-PHB2 antibodies. (**d**) Dephosphorylation of full-length PHB2 phosphorylation at S39 by PP1Cα using Phos-tag SDS–PAGE and western blot analysis with anti-PHB2 and anti-pS39-PHB2 antibodies. (**e**) Phosphatase activity of PP1Cα against purified, recombinant full-length PHB2 phosphorylation at S39. These data represent the means ± s.e.m. of three independent experiments. (**f**) The effects of siPKA and siPKCα on BIG3–PHB2 complexes and PHB2 S39-phosphorylation in MCF-7 cells after E2 stimulation for 6 h (left), 12 h (middle) and 24 h (right). ***P < 0.001 (two-sided Student's t-test).

phosphorylated the PHB2-S39A peptide (Fig. 3b). We also obtained direct evidence that the quadrupole time-of-flight spectra of the precursor fragment with a *m/z* ratio of 683.3 corresponded to the PHB2-WT peptide, which resulted from S39 phosphorylation by PKCα (Supplementary Fig. 4c).

We examined the direct S39-phosphorylation of recombinant full-length PHB2 protein by PKCα using Phos-tag SDS–polyacrylamide gel electrophoresis (PAGE) technology and subsequent western blot analysis with anti-PHB2 and anti-S39-phospho-PHB2 antibodies. Two forms of PHB2, corresponding to unphosphorylated and monophosphorylated PHB2, were clearly detected by Phos-tag SDS–PAGE and western blot analysis (Fig. 3c). The S39-monophosphorylation of PHB2 by PKCα was saturated at the molar ratio of 1:2 (PHB2: PKCα) (Supplementary Fig. 4d). Furthermore, to investigate PP1Cα activity upon the purified full-length PHB2 with phosphorylation at S39 (see Methods), we performed Phos-tag SDS–PAGE and an *in vitro* malachite green phosphatase assay. As expected, PP1Cα dephosphorylated the phosphorylated PHB2 (Fig. 3d), and remarkably increased the amount of free phosphate produced from phosphorylated PHB2 (Fig. 3e). We confirmed a similar dephosphorylation of PP1Cα against the full-length PHB2 with phosphorylation at S39 purified from immunoprecipitation with the phospho-specific PHB2 (S39) antibody in MCF-7 cells treated with E2/ERAP for 24 h (Supplementary Fig. 4e). Therefore, this suggests that PKCα is responsible for E2-induced PHB2-S39 phosphorylation in breast cancer cells.

Next, to solve the conflict that E2 treatment caused inactivation of PHB2 via BIG3-PKA-PP1Cα, but led to activation of PHB2 via PKCα, we examined the impact of PKA and PKC knockdown on the S39-phosphorylation level of PHB2 and the interaction of BIG3 with PKCα, PKA, PHB2 and ERα in the presence of E2 in MCF-7 cells. Our results showed that PKA depletion resulted in the recovery of E2-induced PHB2 S39-phosphorylation, which was immunoprecipitated with BIG3, and the weak interaction of BIG3 with PKCα for 12 h after E2 treatment (Fig. 3f). On the other hand, PKCα depletion did not affect the S39-phosphorylation of PHB2 or the BIG3-PKA and BIG3–PHB2 interaction (Figs 2f,3f; Supplementary Fig. 4f). Notably, we observed no interaction of BIG3 with ERα via PHB2 despite E2 stimulation in PKA- or PKCα-depleted MCF-7 cells, as well as siEGFP-treated cells (Fig. 3f). Therefore, BIG3–PKA–PP1Cα tri-complexes preferentially sustain PHB2 in a dephosphorylated, inactive state even for E2 stimulation, suggesting the possibility that BIG3–PKA–PP1Cα tri-complexes inhibit the approach of ERα and PKCα.

**Dephospho-PHB2-S39 has a poor prognosis**. Previous studies have shown that *BIG3* is frequently upregulated in ERα-positive breast cancers[5,6] and here we found that the overexpression of *BIG3* was significantly correlated with the poorer prognosis of patients with ERα-positive breast cancer based on the RNAseq2 data set from the Cancer Genome Atlas (Supplementary Fig. 5a, Log rank test; *P* < 0.001). To investigate the clinicopathological significance of the association between BIG3 expression and PHB2-S39 phosphorylation levels in breast cancer, we performed immunohistochemistry analyses of 82 primary ERα-positive breast cancer specimens and classified these into four levels of BIG3 expression (score: 0 to +3; see Methods) (Fig. 4a; Supplementary Table 2).

We observed strong, moderate, weak and negative staining in 16, 32, 24 and 10 cases, respectively, indicating an association between strong staining and low survival (two-sided Student's *t*-test; *P* < 0.001; Supplementary Fig. 5b). Kaplan–Meier analysis showed that patients with strong/moderate BIG3 staining had

significantly shorter disease-free survival rates than those with negative/weak BIG3 expression (*P* = 0.004 by the log-rank test; Fig. 4a). Conversely, we observed that high levels of PHB2-S39 phosphorylation were strongly associated with a good prognosis, suggesting that BIG3 contributes to the dephosphorylation of PHB2 in breast carcinogenesis and development (Fig. 4b; Supplementary Fig. 5c). A multivariate analysis indicated that strong BIG3 expression was independently correlated with the poor prognosis of patients with ERα-positive breast cancer and the disease stage (*P* = 0.018; Fig. 4c). More importantly, we observed a significant inverse correlation between BIG3 overexpression and PHB2-S39 phosphorylation levels (Fisher's exact test; *P* < 0.001; Fig. 4d). These findings suggest that BIG3 overexpression and PHB2-S39 dephosphorylation were predictive of worse outcomes in ERα-positive breast cancer patients.

**Discussion**

AKAPs facilitate signalling transduction by tethering signalling enzymes, such as kinases or phosphatases, to their substrates[12,19]. In the present study, we demonstrated for the first time that BIG3 forms a tri-complex with upstream kinase PKA and a regulatory subunit of PP1Cα to function as a novel cancer-specific AKAP, which is classified as a *PPP1R* targeting subunit. In ERα-positive breast cancer cells, E2 stimulation induced PKA-dependent S305 and S1208 phosphorylation in BIG3, which cancelled its negative regulation of PP1Cα activity. This resulted in the apparent inactivation of PHB2 through S39 dephosphorylation via interference of the interaction of PKCα with the BIG3–PKA–PP1Cα complex, likely due to conformation changes in the PKCα-binding region(s) of PHB2 (Fig. 5, upper panel). By contrast, treatment with ERAP resulted in PHB2 being released from BIG3, leading to E2-induced PKCα-mediated PHB2-S39 phosphorylation and the complete suppression of E2 signalling pathways (Fig. 5, lower panel). Of note, the non-phosphorylated form of PHB2 (such as S39A) also interacts with ERα, but does not repress signalling pathways. Moreover, the depletion of PKA or PP1Cα also resulted in E2-induced PHB2-S39 phosphorylation (Fig. 2f), suggesting that BIG3 may act as an anchoring protein for PKA and PP1Cα to maintain PHB2 in a dephosphorylated, inactive state. Further analyses are required to elucidate whether the phosphorylation site overlaps with the binding region of ERAP for release from BIG3 (ref. 6) or karyopherin for the nuclear translocation of PHB2 (ref. 9). In contrast to these findings, previous studies have provided compelling evidence that increased PKCα expression leads to a more aggressive phenotype[20,21] and is associated with resistance to cytostatic drugs in breast cancer cell lines[22,23]. This discrepancy may reflect the fact that the PKCα targeting of PHB2 for tumour suppression may be non-functional by PKA–BIG3–PP1Cα complex, thereby inducing an apparent 'loss-of-function' of the PHB2 protein.

This study provides a novel insight into the mechanism of E2 action, whereby E2 stimulates PKA, which binds to BIG3, thereby phosphorylating BIG3-S305 and -S1208, and cancelling the inhibitory effect of BIG3 against PP1Cα activity. This suggests that BIG3 may act as a coordinator for the constitutive activation of E2-ERα signalling in breast cancer cells. We used *in silico* analysis[24] to identify two potential regions with PKA in BIG3 (318–338 and 1,225–1,245 amino acids; Supplementary Fig. 1g), the first of which is close to Domain B in BIG2 (285–305), which has been experimentally verified as a RIIBD[10]. This finding is reasonable, reflecting the spatial proximity to two PKA phosphorylation sites (S305 and S1208) in BIG3. Moreover, a growing body of evidence has revealed that E2 stimulation increases intracellular cAMP, thereby enhancing PKA activity in breast cancer cells[25,26]; and that G-protein-coupled oestrogen

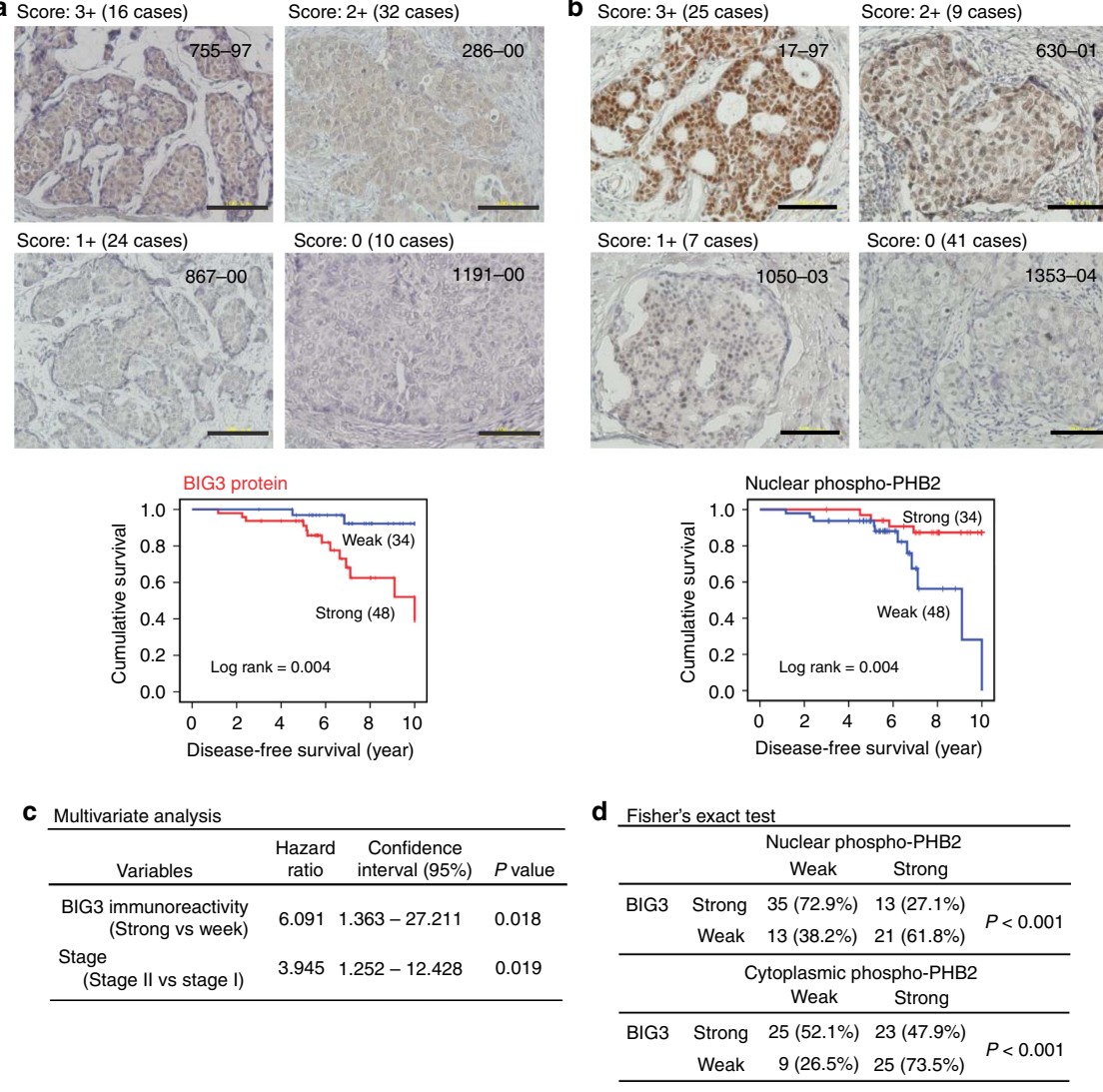

**Figure 4 | BIG3 overexpression and PHB2 dephosphorylation status are strongly associated with poor prognosis.** (**a,b**) Immunohistochemical staining (upper) and Kaplan–Meier analysis (lower) of survival associated with BIG3 protein (**a**) and nuclear PHB2 phosphorylation at S39 (**b**) in representative ERα-positive breast cancer specimens. (**c**) BIG3 protein expression, showing an independent prognostic significance in the multivariate Cox regression analysis. (**d**) Reverse correlation of BIG3 expression with PHB2 phosphorylation status at S39 according to Fisher's exact test. For overall survival analysis, the log-rank (Mantel-Cox) test was used to determine the statistical difference between stratified groups.

receptor (E2-GPR30), which is potentially involved in breast carcinogenesis[27–31], initiates a rapid response through adenylyl cyclase activation and cAMP accumulation, which further leads to PKA activation[25]. PKA overexpression is also associated with high proliferation in the normal breast, malignant transformation in the breast, a poor prognosis in established breast cancer and resistance to anti-oestrogens[32–34]. Consequently, there is strong evidence that E2-stimulated cAMP likely enhances PP1Cα activity via PKA, which binds to BIG3, resulting in the dephosphorylation of PHB2-S39, suggesting that BIG3 acts as an AKAP for E2 signalling activation in luminal breast cancer. However, further analyses are required to elucidate the mechanism of cAMP upregulation by E2 stimulation.

Although the mechanism by which the PKA-mediated phosphorylation of BIG3-S305 and -S1208 cancels the inhibitory activity of BIG3 remains unclear, recent structural studies of phosphatase inhibitor proteins may provide some clues. For example, it has been found that the phosphorylation at T72 of Inhibitor-2 (I-2), one of the most ancient PP1 inhibitor, by glycogen synthase kinase-3 forms an inactivate complex with PP1 which can be reactivated[35]. The cocrystallization of PP1 with I-2 shows three critical binding regions containing an RVXF-binding motif, a SILK interaction site and a long alpha-helix loop including T72 phosphorylation site[36]. The T72-phosphorylation on I-2 has been shown to increase the fluorescence anisotropy of a label covalently linked to a cysteine engineered into residue 129 of I-2, indicating a conformational change in PP1;I-2 complex formation upon T72 phosphorylation[37]. Here we demonstrated that both S305A and S1208A mutants of BIG3 still bound to PP1Cα (Fig. 1f) due to the remaining intact RVxF motif, indicating that the E2-induced PKA-mediated phosphorylation of BIG3-S305 and -S1208 may result in a conformational change, exposing an inhibitory surface of PP1Cα. Alternatively, other PP1-regulatory/targeting holoenzymes are known to induce to auto-inhibition in response to the phosphorylations of holoenzymes themselves[38–41], suggesting the possibility that BIG3 may possess the potential auto-inhibitory sites for PP1Cα activation. Additional structural/functional studies are required to elucidate the precise mechanism whereby BIG3 phosphorylation inhibits PP1Cα activity.

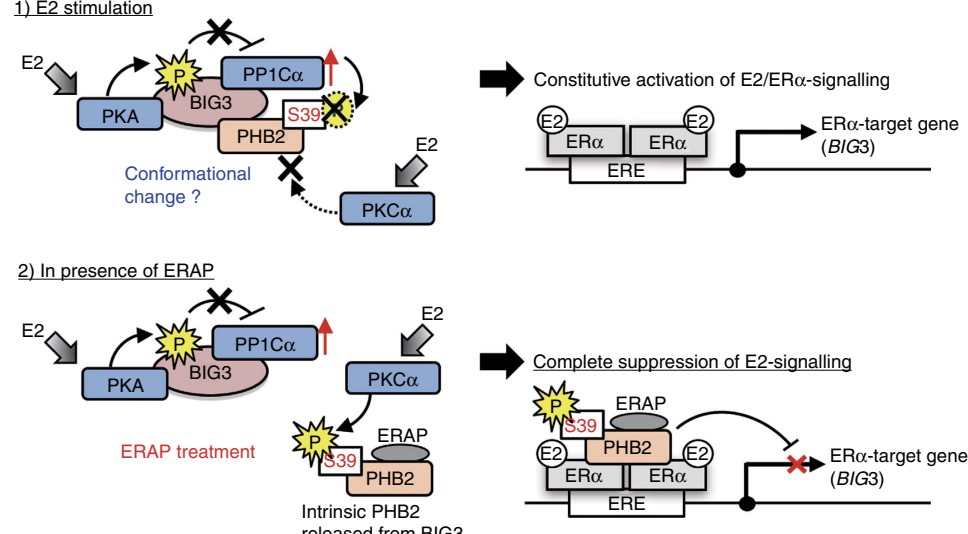

**Figure 5 | Schematic illustration of the novel mechanism of PHB2 inactivation through the BIG3–PKA–PP1Cα tri-complex, in breast cancer cells.** Upper, E2 stimulation induces PKA-dependent BIG3 phosphorylation, which cancels its negative regulation of PP1Cα activity, resulting in the avoidance of PHB2 suppressive activity and PKCα approach. Lower, ERAP treatment induces PHB2 release from BIG3, continuously inducing PHB2 phosphorylation via PKCα and resulting in the complete suppression of ERα signalling pathways.

Immunohistochemical clinicopathological evidence revealed that high BIG3 expression levels and low PHB2 phosphorylation levels are strongly associated with a poor prognosis in patients with ERα-positive breast cancers. Notably, 12 of 15 recurrent specimens exhibited strongly positive BIG3 staining and almost undetectable PHB2 phosphorylation, indicating a possible association between strong BIG3 expression and prognostic recurrence. However, several cases showed no or weak BIG3 expression but no or weak PHB2 phosphorylation (Supplementary Table 2). This discrepancy may reflect the loss of functional PKC mutations in these cases, as PKC is frequently mutated in human cancers[42].

Collectively, BIG3–PKA–PP1Cα tri-complexes preferentially inactive PHB2 via PHB2-S39 dephosphorylation in the presence of E2 stimulation. These findings indicate the precise pathophysiological contribution of BIG3 to PKA and PP1Cα for the inactivation of PHB2, which exerts a highly potent suppressive activity against multiple E2/ERα signalling pathways in E2-dependent breast carcinogenesis. A better understanding of the precise pathophysiological roles of BIG3 in ERα-positive breast cancer may provide new approaches for therapeutic or diagnostic interventions in future preclinical and clinical studies.

## Methods

**Ethical statement.** All experiments were conducted according to protocols reviewed and approved by the Committee for Safe Handling of Living Modified Organisms in Tokushima University (Permission number 28–5).

**Generation of antibodies.** BIG3 monoclonal antibody (459–572 amino acid as an antigen) for western blot analysis and immunoprecipitation was obtained from Sigma-Aldrich as previously described[7]. Phospho-BIG3 (S305), phospho-BIG3 (S1208), and phospho-PHB2 (S39) were obtained from Scrum. Each antibody was a recognized phosphorylated peptide, namely, SDHGRG(pS)GCS, RCW(pS)LVAPH, and YGVRE(pS)VFTVE, respectively. For immunohistochemical analyses, the anti-BIG3 polyclonal antibody was obtained from Medical and Biological Laboratories. Briefly, His-tagged human BIG3 recombinant protein (799–1,200 aa) was purified using Ni-NTA resin agarose (Qiagen), and immunized into rabbits. The immune sera were then purifies on antigen affinity columns.

**Cell lines and culture conditions.** Human breast cancer cell lines (MCF-7, ZR-75-1, BT-474, T47D and HCC1500) and HEK293T cells were obtained from American Type Culture Collection. HBC4 was kindly gifted from Dr Takao Yamori (Cancer Chemotherapy Centre, Japanese Foundation for Cancer Research). The KPL-1 and KPL-3C cells were established, characterized[43,44], and kindly gifted from Dr Jun-ichi Kurebayashi (Kawasaki Medical School). All cell lines were cultured under the respective depositors' recommendations. The cell lines stocks that were used in this study had been properly stored in liquid nitrogen. We monitored the cell morphology of these cell lines by microscopy and confirmed that they maintained their morphology by comparing images with the original morphologic images. No *Mycoplasma* contamination was detected in any of the cultures using a *Mycoplasma* Detection kit (Roche).

MCF-7 cells were seeded onto 24-well plates ($1 \times 10^5$), 6-well plates ($5 \times 10^5$), or 10-cm dishes ($2 \times 10^6$) in MEM (Thermo Fisher Scientific) supplemented with 10% FBS (Nichirei Biosciences), 1% antibiotic/antimycotic solution (Thermo Fisher Scientific), 0.1 mM NEAA (Thermo Fisher Scientific), 1 mM sodium pyruvate (Thermo Fisher Scientific) and 10 μg ml$^{-1}$ insulin (Sigma). The cells were maintained at 37 °C with 5% $CO_2$. The next day, the medium was changed to phenol red-free DMEM/F12 (Thermo Fisher Scientific), supplemented with FBS, antibiotic/antimycotic solution, NEAA, sodium pyruvate, and insulin. After 24 h, the cells were treated with 10 nM of 17β-oestradiol (E2, Sigma E8875).

**Inhibitors.** The dominant-negative peptide inhibitors, ERAP (RRRRRRRRRRR-G GG-QMLSDLTLQLRQR), contained the BIG3-binding resides, and BIG3/PP1Cα binding peptide inhibitor (RRRRRRRRRRR-GGG-QKAVSFIHDILTEVL) contained the PP1Cα-binding motif, were synthesized to specifically inhibit the BIG3–PHB2 and BIG3–PP1Cα interaction, respectively. H-89 (Sigma, B1427) and okadaic acid (Sigma, O9381) were used for inhibitor of PKA and PP1Cα, respectively. MCF-7 or KPL-3C cells were treated with 10 nM E2 ± ERAP, BIG3/PP1Cα inhibitor, H-89, or okadaic acid for 24 h.

**Specimens.** Surgically resected breast cancer tissue samples and their corresponding clinical information were acquired from the Tokushima Breast Care Clinic (Tokushima, Japan) and the National Hospital Organization Higashitokushima Medical Centre after obtaining written informed consent (Supplementary Table 2). The present study and the use of all clinical materials described above were approved by the Ethical Committee of Tokushima University (Permission number #28–16).

**Immunoblot analyses.** The cells were lysed with lysis buffer (50 mM Tris-HCl; pH 8.0, 150 mM NaCl, 0.1% NP-40 and 0.5% CHAPS) containing 0.1% protease inhibitor cocktail III (Calbiochem, 539134). The lysates were electrophoretically separated, blotted onto a nitrocellulose membrane, and blocked with 4% BlockAce solution (Dainippon Pharmaceutical, UK-B80) for 1 h. The blots were subsequently incubated with antibodies against the following proteins: BIG3 (1:1,000, ref. 7), phospho-BIG3 (S305, 1:500); phospho-BIG3 (S1208, 1:500); and phospho-PHB2 (S39, 1:500); PHB2 (1:500) (Abcam, ab15019); PKAα cat (C-20, 1:1,000, sc-903), PKCα (H-7, 1:500, sc-8393) and PP1Cα (FL-18, 1:500, sc-443) (Santa Cruz Biotechnology); ERα (SP1, 1:500, RM-9101-S0), phosphoserine (1:1,000, 61–8100), and phosphothreonine (1:1,000, 71–8200) (Thermo Fisher Scientific); IGF-1Rβ (1:500, #3027), phospho-IGF-1Rβ (Tyr1131, 1:500, #3021S) and α/β-tubulin

(1:1,000, #2148) (Cell Signalling Technology); β-actin (AC-15; 1:5,000, A5441), LMNB1 (1:100, SAB1400153), and FLAG-tag M2 (1:5,000, F3165) (Sigma); and HA-tag (3F10,1:3,000; Roche, 1867423). After incubation with an horseradish peroxidase-conjugated secondary antibody (Santa Cruz Biotechnology, dilution 1:5,000, rabbit; sc-2004, mouse; sc-2005, rat; sc-2006) for 1 h, the blots were developed using an enhanced chemiluminescence system (GE Healthcare, RPN2106) and scanned using an Image Reader LAS-3000 mini (Fujifim). All experiments were performed more than two times in duplicate. Full-length images of immunoblots are shown in Supplementary Fig. 6.

**Phos-tag SDS–PAGE.** To analyse PHB2 phosphorylation, phos-tag SDS–PAGE was performed using precast SuperSep gels (50 μM phos-tag acrylamide and 100 μM ZnCl$_2$, Wako Chemicals, 190–16721), as previously described with some modification[45]. Molecular size markers for phos-tag SDS–PAGE used the WIDE-VIEW Prestained Protein Size Marker III (Wako Chemicals, 236–02463). The percentage of phosphorylation was calculated according to the formula: (phosphorylated PHB2 band area/total PHB2 band area) × 100.

**Immunoprecipitation.** The cells were lysed with 0.1% NP-40 lysis buffer as described above. The cell lysates were pre-cleared with normal IgG and rec-Protein G Sepharose 4B (Thermo Fisher Scientific, 101242) at 4 °C for 3 h. Subsequently, the supernatants were incubated with antibodies against BIG3 (5 μg), PHB2 (5 μg), PP1Cα (5 μg), IGF-1Rβ (5 μg) and IgG (5 μg) at 4 °C for 12 h. Next, the antigen–antibody complexes were precipitated using rec-Protein G Sepharose 4B at 4 °C for 1 h. Immunoprecipitation of FLAG was conducted using anti-FLAG M2 affinity gel (Sigma, A2220). Immunoprecipitated protein complexes were washed three times with lysis buffer and separated using SDS–PAGE. Immunoblot analyses were performed as described above.

**Purification of PHB2 phosphorylation at S39.** After incubation of recombinant PHB2 (Abnova, H00011331-P01) with recombinant PKCα (Merck Millipore, 14–484), PKC lipid activator (Merck Millipore, 20–133) and 0.5 mM ATP (Takara, 4041) for 30 min at 30 °C, the reaction samples were incubated with anti-phospho-PHB2 (S39) antibody at 4 °C for 12 h. The complexes were then precipitated using rec-Protein G Sepharose 4B at 4 °C for 1 h. After washing three times with lysis buffer, the purification of full-length PHB2 with phosphorylation at S39 was verified using immunoblot analysis.

**Phosphatase assay.** The PP1Cα phosphatase activity was determined using the *p*NPP protein phosphatase assay kit (AnaSpec, AS-71105), according to the manufacturer's instructions. Briefly, 50 μl of the samples and 50 μl of *p*NPP were added to the wells of a 96-well plate, and the colour was developed for 60 min. After terminating the enzyme reaction, the absorbance at 415 nm was measured using a microplate reader (Infinite M200, Tecan). The PP1Cα activity was defined as the amount of PP1Cα that catalysed the conversion of 1 μmole *p*NPP per minute.

**Dephosphorylation assays of BIG3 using malachite green.** HEK293T cells were transfected with FLAG-tagged BIG3 constructs (wild-type and S1208A) and HA-tagged ERα constructs. After 24 h, the cells were treated with E2 for 24 h, and the lysates were immunoprecipitated using an anti-FLAG M2 affinity gel. The PP1Cα activities of the immunoprecipitates for phosphorylated PHB2 were determined with the Ser/Thr Phosphatase Assay Kit 1 (Merck Millipore, 17–127). Briefly, the immunoprecipitates were added to PP1Cα reaction buffer (20 mM Tris-HCl; pH 7.4, 0.01% Brij 35) containing 15 mg ml$^{-1}$ of phospho-PHB2 peptide (YGVRE(pS)VFTVE). After 30 min, the supernatants were collected and mixed with malachite green solution for 10 min. The absorbance of malachite green at 620 nm was defined as phosphatase activity that releases phosphates from the phospho-PHB2 peptide.

**In vitro kinase assay.** PKA and PKCα activity was determined using the ADP-Glo kinase assay Kit (Promega, V6930), according to the manufacturer's instructions. Briefly, after the kinase reaction (300 ng recombinant PKA, 5 μg BIG3 peptide and ATP for PKA activity; 100 ng PKCα, PKC lipid activator, ATP for PKCα activity) at 30 °C for 30 min, ADP-Glo reagent was added to terminate the kinase reaction and deplete the remaining ATP, and the kinase detection reagent was added to convert ADP to ATP and facilitate the measurement of the newly synthesized ATP using a luciferase/luciferin reaction. The light generated correlates with the amount of ADP generated in the kinase assay, indicative of kinase activity.

**Mass spectrometry analysis.** Phosphorylation or dephosphorylation of PHB2 was used for mass spectrometry analysis. Briefly, 0.5 mM PHB2 peptide (YGVRESVFTVE) was mixed with recombinant PKCα and PKC lipid activator for 30 min at 30 °C. For the dephosphorylation of PHB2, phospho-S39 PHB2 peptide was mixed with recombinant PP1Cα (Merck Millipore, 14–595) for 30 min at 30 °C. After cooling on ice, the samples were further purified using ZipTip C18, and subsequent Q-Tof analysis was performed using Q-Tof Ultima API

(Waters-Micromass). The mass spectrometry (MS) scan was performed between *m/z* 50–1,500 (MS) and 50–1,500 (MS/MS). The obtained MS and MS/MS data were analysed using the MASCOT server programme (version 2.0.05; Matrix Science). PHB2 peptide and phospho-S39 PHB2 peptide were synthesized by Sigma.

**2DICAL analyses.** Phosphorylation analysis of BIG3 was performed using 2DICAL (two-dimensional image-converted analysis of liquid chromatography and mass spectrometry)[46]. Briefly, 0.5 mM BIG3 peptide (S305; DHGRGSGCSCTAPALSGPVARVAPH and S1208; RCWSLVAPH) was mixed with 10 ng μl$^{-1}$ recombinant PKA (Merck Millipore, 539482) and 0.5 mM ATP for 30 min at 30 °C. After cooling on ice, the reaction samples were mixed with 5 M urea, 1 M NH$_4$HCO$_3$, and Sequencing Grade Modified Trypsin (Promega, V5113). After a 20- h digestion at 37 °C, 5% formic acid was added, followed by centrifugation at 20,000 *g*. The supernatants were incubated with an equal volume of ethyl acetate and subsequently centrifuged at 20,000 *g*. The peptides were collected from the lower layer, dried with EYELA CVE-3100 and UT-1000 (Tokyo Rikakikai), and dissolved in 0.1% formic acid. Liquid chromatography–mass spectrometry and data acquisition were performed as previously described[44].

**Immunohistochemical staining.** To examine BIG3 expression and the phosphorylation of cytosolic or nuclear PHB2 at S39 in ERα-positive breast cancer tissues, we stained 3-μm sections of paraffin-embedded tumours with polyclonal anti-BIG3 (dilution 1:200) or anti-phospho-PHB2 (S39, dilution 1:100) antibodies, as previously described[47]. Experienced pathologists without prior knowledge of clinicopathological data semi-quantitatively evaluated the intensity of BIG3 or PHB2 phosphorylation staining based on the following scoring system: 0, negative (no staining); 1, weak; 2, moderate; and 3, positive. The staining scores were divided into the following 2 groups: weak (scores 0 and 1) and strong (scores 2 and 3).

**Construction of BIG3 and PHB2 expression vectors.** To construct the expression vectors, each coding sequence was PCR-amplified using KOD-Plus DNA polymerase (Toyobo, KOD-201). The PCR product for BIG3 was inserted into the *Eco*RI and *Not*I sites of the pCAGGSn3FH expression vector in frame with a FLAG tag at the N terminus and a haemagglutinin (HA) tag at the C terminus. For BIG3 mutant (S305A-S1208A), BIG3 mutant (S305A) and BIG3 mutant (S1208A) were digested by *Eco*RI-*Xho*I and *Xho*I-*Not*I, respectively, and then each partial construct inserted into the pCAGGSn3FH expression vector. The PHB2 expression vectors was inserted in frame into the *Eco*RI and *Xho*I sites of the pCAGGSnHC expression vector with an HA tag at the C terminus. The sequences of each primer set are described in Supplementary Table 3. The DNA sequences of the constructs were confirmed by DNA sequencing (ABI3500xL x24).

**Real-time PCR.** Expression of the *PRKCA* (PKCα), *PRKCE* (PKCε), and *CAMK2B* (CAMK2) genes was evaluated by real-time reverse transcription PCR. The extraction of total RNA and subsequent complementary DNA synthesis were performed. The complementary DNA was analysed in duplicate by real-time PCR using a 7500 Real Time PCR System (Applied Biosystems) and SYBR Premix Ex Taq (Applied Biosystems, 4367659) according to the manufacturer's instructions. Each sample was normalized to *β2-MG* messenger RNA content. The data represent the means ± s.e.m. of three independent experiments. The sequences of each primer set are described in Supplementary Table 3.

**ChIP assay.** ChIP analysis was performed using the EZ-ChIP system (Merck Millipore, 17–371) according to the manufacturer's instructions. Briefly, MCF-7 cells were treated with E2 for 24 h, and then fixed in 37% formaldehyde, resuspended in lysis buffer, and sonicated 10 × 10 s (amplitude 5%) in a Microson XL-2000 sonicator (Misonix). Supernatants were pre-cleared with Protein G agarose beads and 1% of the input was reserved for later analysis. Lysates (from 1 × 10$^6$ cells for each sample) were then immunoprecipitated overnight at 4 °C using antibodies to ERα or normal mouse IgG. DNA–protein complexes were captured by incubation for 1 h at 4 °C with Protein G agarose beads. Complexes were washed, resuspended in elution buffer and incubated for 5 h at 65 °C to reverse cross-links. DNA was then isolated, and EREs were detected with PCR. Primers were as follows: ERE ( − 726 to − 704 bp from the transcriptional start site) in the 5′ upstream region of *PPP1CA*; 5′-TCAAAAGCTAATTATGGG GCC-3′ and 5′-TCAAGCGATTCTCCTGCCTCA-3′.

**Luciferase reporter assay.** HEK293T cells were transfected with an ERE (SABiosciences, CCS-005L), FLAG-tagged ERα and HA-tagged PHB2 (WT) construct or PHB2 mutant construct (S39A). pRL-TK was monitored as an internal control. At 16 h post-transfection, the culture medium was changed to assay medium (10% FBS in Opti-MEM). After 8 h, the cells were exposed to E2 for 24 h. Subsequently, the cells were harvested and analysed for luciferase and *Renilla*-luciferase activities using the Promega dual luciferase reporter assay (Promega, E1910). All data were normalized to *Renilla*-luciferase to account for transfection

efficiency. The data are expressed as the fold increase over untreated cells (set at 1.0) and represent the means ± s.d. of three independent experiments.

For E2-dependent direct transactivation of *PPP1CA* by ERα, MCF-7 cells were individually transfected with a luciferase reporter containing an ERE motif conserved within the 5′ upstream region of the *PPP1CA* gene (GGCCACCTGGCCA).

**Nuclear/cytoplasmic fractionation.** Nuclear and cytoplasmic/plasma membrane fractions were prepared using the NE-PER nuclear and cytoplasmic extraction reagent (Thermo Fisher Scientific, 78835) according to the manufacturer's instructions. α/β-Tubulin and laminin B were used as loading controls for the cytoplasmic and nuclear fractions, respectively.

**PKA-binding site prediction.** PKA-binding sites in BIG3 were predicted using the method of Hou *et al.*[24] for detecting potential RIIBDs. The method identifies candidate motifs that must satisfy the following conditions: (1) The peptide sequence consists of 21 amino acids and matches the motif of: [AEGILMQSTV]-{P}-{P}-{P}-[AFGILMSTVY]-{P}-{P}-[FILV]-[AILSTV]-{P}-{P}-[AILSTVQ]-[ILSTV]-x-x-[AFGILSTV]-{P}-{P}-{P}-x-x, where {P} means not a proline, x means any residue, [FILV] means F, I, L or V. (2) The sequence produces a good matching score based on the position-specific score matrix described in Hou *et al.*[24] The smaller values in Supplementary Fig. 1g indicate better matches. (3) The predicted secondary structure is alpha-helical. We used PSI-PRED[48] for predicting secondary structures. (4) The sequence contains no trans-membrane regions. We used TMHMM[49] and HMMTOP[50] for predicting trans-membrane regions. (5) The sequence is conserved among different species. The conservation was assessed by comparing the human, mouse, chimpanzee, monkey, cow, dog and rat sequences with PAM500, as described in Hou *et al.*[24] In addition, we also used PSIVER[51], a method to predict residues that bind to other proteins, to confirm that the predicted RIIBDs included potential protein-binding sites. The default threshold of 0.390 was used in the present study.

**TCGA data set for BIG3 expression.** BIG3 expression and the survival analysis were evaluated using human breast tumour from publicly available The Cancer Genome Atlas (TCGA) RNA-seq 2 data set. Seventy-three samples of ERα-positive breast cancer with a matched normal sample were selected, and compared using two-tailed Student's *t*-test. For disease-free survival analysis, the BIG3 expression intensity was divided into high (26 cases) and low (47 cases) based on median value, and statistical differences determined by the log-rank (Mantel-Cox) test or Kaplan–Meier analysis.

**Statistical analyses.** Progression-free survival curves were estimated using the Kaplan–Meier method. The statistical significance of a relationship between the clinical outcome and BIG3 expression or the phosphorylation of cytosolic or nuclear PHB2 at S39 was assessed using the trend log-rank test. Cox proportional hazards analysis was used to identify significant prognostic clinical factors and to test for an independent contribution of BIG3 expression to progression-free survival. The significance of differences between the experimental groups was calculated using Student's *t* test. A difference of $P < 0.05$ was considered statistically significant.

**Data availability.** Phosphorylation site prediction in Supplementary Information Fig. 1k is available from NetPhos 3.1 Server (http://www.cbs.dtu.dk/services/NetPhos/). We used PSI-PRED (http://bioinf.cs.ucl.ac.uk/psipred/) for predicting secondary structures, TMHMM (http://www.cbs.dtu.dk/services/TMHMM/) and HMMTOP (http://www.enzim.hu/hmmtop/) for predicting trans-membrane regions, and PSIVER (http://mizuguchilab.org/PSIVER/) for predicting protein–protein interaction sites in protein sequences. The TCGA data set referenced in Supplementary Information Fig. 5a were assessed from TCGA portal (http://cancergenome.nih.gov/). All the other data supporting the findings of this study are available within the article and its Supplementary Information Files or from the corresponding authors upon reasonable request.

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

## Acknowledgements

We thank Dr Takao Yamori (Cancer Chemotherapy Centre, Japanese Foundation for Cancer Research) for gifting the HBC4 breast cancer cell line, Dr Junichi Kurebayashi (Kawasaki Medical School) for gifting the KPL-3C breast cancer cell lines and Ms Hinako Koseki and Ms. Hitomi Kawakami for providing excellent technical support. We would like to thank Enago (www.enago.jp) for the English language review. This work was financially supported by a grant/research support from Tokushima Breast Care Clinic, Grants-in-Aid for Scientific Research (B) (MEXT KAKENHI Grant Number 25293079 and16674279) and (C) (MEXT KAKENHI Grant Number 26461948), and Grants-in-Aid for Scientific Research on Innovative Areas (MEXT KAKENHI Grant Number 16701519).

## Author contributions

T.Y. performed all the experiments. M.O. performed the 2DICAL analysis. Y.B. evaluated the immunohistochemial results. Y.-A.C. and K.M. performed in silico analyses for the prediction of PKA binding sites. H.S. provided the protocols and techniques for the measurement of phosphatase activity. M.K. performed functional analyses concerning BIG3-PKA-PP1Cα. I.I. performed statistical analyses of the immunohistochemical results. K.I. evaluated the immunohistochemial results. Y.M. discussed the interpretation of the BIG3 effect on ERα-signalling pathway data. M.S. and J.H. collected breast cancer specimens. T.K. was involved in the conception and design of all studies, interpretation of the data, and preparation of the draft and final version of the manuscript. All authors read and approved the final manuscript.
