## [Peer Review File · Nature Communications]

Reviewers' comments:

Reviewer #1 (Remarks to the Author):

The novelty of this submission is predicated upon the discovery that the mechanism regulating Brefeldin A-inhibited guanine nucleotide-exchange protein 3 (ARFGEF3; herein termed BIG3) in estrogen signaling of ERa-positive breast cancer cells involves novel coordination of PKA and PP1 activities. Namely, the submission provides biochemical and functional evidence to support:

- 1) BIG3 acting as both an A-kinase-anchoring protein (AKAP) and protein phosphatase regulatory/binding subunit (PPP1R).
- 2) PKA phosphorylation of BIG3 at S305 and S1208 acting to inhibit the phosphatase holoenzyme (although the mechanism remains undefined).
- 3) BIG3 impacting upon the PKCa-dependent phosphorylation and nuclear translocation of prohibitin-2 (PHB2: a repressor of estrogen receptor activity).
- 4) Furthermore, the study provides novel insight in linkages of BIG3 and pS39-PHB2 levels in ERa-positive breast cancer specimens.

Overall, this was a logically executed study. For the most part, there are appropriate controls and experimental justification of conclusions. Although the compartmentalization of PKA and PP1C signaling molecules on a single targeting/regulatory protein is not novel, I would expect the study to be of keen interest to other researchers in the PP1C phosphatase field and the ERa-cancer biology field.

PRIMARY CONCERNS:

A concern is that the PP1C binding property of BIG3 was already identified in a proteomic survey completed by Mattieu Bollen's group (Chem Biol 2009; 16, 365-71). Indeed the protein is already annotated by the HGNC to be a member of the protein phosphatase regulatory subunit family (designated PPP1R33; <http://www.genenames.org/cgi-bin/genefamilies/set/694>). However, this proteomic survey performed by Bollen's group on the PP1C binding proteins did not interrogate the functional impact of BIG3 as a PPP1R. So, the identification of AKAP and PPP1R properties for BIG3 (along with the influences on downstream signaling) in this study should be considered impactful to the field of ERa-signalling in cancer biology. I would suggest that the authors acknowledge that BIG3 is also annotated as PPP1R33.

In Fig1B, immunoprecipitations of PP1Ca were completed, and phosphatase activity was measured against pNPP (non-specific substrate). There are no differences in PP1Ca amounts in pull-down, so it is unclear why the authors observe differences in PP1Ca activity. They are actually monitoring total PP1Ca activity against pNPP in the pull-down; the experimental design will not select for phosphatase activity specific to the BIG3-PP1Ca complex. Also there appears to be significant BIG3 pull-down in the absence of any PP1Ca. Is this immunoreactivity against a non-specific target? The authors should perform the co-IP of BIG3 protein to provide additional substantiation for the presence of an endogenous complex. It would be prudent for the authors to use a substrate that is specific for the BIG3:PP1Ca holophosphatase.

In Fig1f, all the IPs from E2 stimulated HEK293T cells have significantly more PP1Ca even though similar amounts of FLAG-BIG3 are present in the pull downs. It is not clear what substrate was employed (presumably pNPP, non-specific phosphatase substrate) or if the authors are normalizing this activity against the PP1Ca signal recovered in the IP. It would be prudent for the authors to use a substrate that is specific for the BIG3:PP1Ca holophosphatase.

In Fig 2c-e, why the switch to HEK293T cell system? Is this because the molecular components of the BIG3-PHB2-ERa signaling cascade are not endogenously expressed?

In Fig 2g, the authors examine PP1Ca activity toward pS39-PHB2 peptide. It is not surprising that PP1Ca will dephosphorylate this peptide in vitro. The experiment does little to offer convincing

evidence that a native BIG3-PP1Ca holophosphatase complex preferentially targets the pS39 site of PHB2.

In Fig3bc, the experimental design lacks rigour. Again it is not surprising that purified PKCa will phosphorylate S39-PHB2 peptide in vitro, so the MS data are not very impactful. The kinase assay employed does not directly assess phosphoryl transfer to the PHB2-peptide substrate. It is unclear from the experiment whether biologically relevant amounts of phosphorylation are observed. To the authors' credit, the follow-up MS experiment does provide validation that the peptide was phosphorylated by PKCa. However, it would be prudent for the authors to provide a true quantitative assessment of PHB2 protein phosphorylation by PKCa (i.e. stoichiometry mol Pi incorporated per mol protein). I expect the PhosTag-SDS-PAGE technology could be easily adapted to this need [References include: Am J Physiol 2016; 310, C681-C91 and Biochem J 2016; 473;2671-85 and Biochim Biophys Acta 2015; 1854,601-8] The aa sequence surrounding the S39 residue possesses only weak similarity to the PKC consensus sequence. The KD/KO of PKCa in cell-based experiments could have confounding effects on other kinases (or phosphatases) that influence S39 phosphorylation status. The use of phosphospecific antibody for pS39-PHB2 is suggestive of its association in nuclear translocation. However, the phosphorylation signal is not quantitative and it is not clear whether a biologically significant level of PHB2 phosphorylation occurs with ERAP treatments. It is possible that additional or different phosphorylation events may be occurring.

MINOR

1. Please include the institutional protocol # for human ethics approval.
2. There is some difference in N-terminal sequence between KIAA1244 (EAW47918 or AAI37175) and the UniProtKB/Swiss-Prot annotated sequence (Q5TH69). The authors should specify the sequence annotation used for the protein biochemistry.
3. Is PKA detected by WIB – catalytic or regulatory subunit. Since similar amounts of PKA are pulled down with IP of BIG3, why are there distinct levels of pSer and pThr events. Are cAMP levels elevated by E2; hence, greater PKA activity?
4. The authors should be careful to distinguish PP1C non-specific activity measures with pNPP vs that measured for the BIG3-PP1C holoenzyme complex.
5. Wording- pg9, line 137-9; how does reaction of S305 and S1208 peptides with recombinant active PKA significantly increase PKA activity?
6. Are other PP1C binding motifs present in BIG3? i.e., G/SILK or MyPhoNe sequences?
7. Pg10, line 144-146. I am supportive of the authors concluding statement that BIG3 forms a heterotrimeric signaling complex with PP1C and PKA; however, the data that support the “enhancement of PP1C activity in the presence of E2 are not as convincing. IP and measurement of BIG3-PP1C phosphatase complex activity against a biologically validated substrate such as PHB2 is required.
8. P11, line 160; spelling “phosphoarylated”
9. P12, line 192-3; “indicating BIG3 phosphorylation enhanced phosphatase activity” Specifically it appears from the data in Fig 2e that both S305 and S1208 phosphorylations are required to achieve maximal PHB2 phosphatase activity.
10. Wording-Pg14, line 216; “the significant activation of the PHB2-WT peptide”. What is the measure of peptide “activity”? Do the authors just mean that PKC can phosphorylate the PHB2 peptide?
11. Were investigators blinded to classification of IHC pathology slides?
12. Pg16, line 257; “a novel cancer-specific AKAP” BIG3 should also be classified as a PPP1R targeting subunit unless the authors feel the dominant influence of BIG3 is on the kinase signaling.
13. Is the targeting function of BIG3 for PP1C and PKA also conserved among the other ARFGEF members (BIG1 and BIG2) or is this a unique architectural feature of the BIG3 protein? Interestingly, the KAVSF (KVxF) motif that provides PP1C-binding appears to be absent in the plant variant of BIG3 (A. thaliana: Q9LPC5.1: although a KVLFD sequence is present at residue 1355).

14. Pg19, line313; "does not have an inhibitory effect" Do the authors mean a disruptive effect on holophosphatase complex stability?

15. Fig2e; pS305-BIG3 mislabelled as "pS1305"

16. Fig3a; Were WCL actually used for loading controls or do the WCLs represent total cytoplasm and nuclear fractions used for the IPs.

Review submitted by: Prof. Justin MacDonald, University of Calgary

Reviewer #2 (Remarks to the Author):

This manuscript dissects an intriguing and intricate signaling mechanism which lies upstream of the estrogen receptor (but perhaps also downstream....; but for sure downstream of the hormone estrogen). The result is the regulation of the inhibitory activity of an ER corepressor. In its phosphorylated form it is a stronger ER repressor, which is good for breast cancer patients because it blunts the proliferative stimulus of ER activity. BIG3 play an unexpected role as a scaffold protein bringing together both a kinase (PKA) and a phosphatase (PP1) as its substrate. Overall, this is a very interesting story with conclusions that are largely very well supported. Considering the complexity of this signaling mechanism, it is not surprising that there are few loose ends that should be tied up.

Major comments:

(1) Connection with ER (see also comment #7): while some consequences for ER activity have been looked at, it is not very clear how the known ER-PHB2 interaction fits into this story. That ERAP promotes the phosphorylation of PHB2 and thereby repression is well documented here, but this is obviously a pharmacological situation. Could it be that ER is also a component of the tri-complex? Perhaps through PKA or even PP1? What's confusing is that the hormone estrogen (E2) both promotes the inactivation of the repressor PHB2 (through PKA) and activates it (through PKC). What happens in the real world (without ERAP) when both pathways are turned on simultaneously by E2? Is it a question of balance? Perhaps playing with that by overexpressing or knocking down the two kinases downstream of E2 might clarify this.

(2) Several IPs lack negative controls: Fig. 1a, Fig. 2d, Extended Data Fig. 1b, Extended Data Fig. 2c.

(3) E2 activation of PP1 seems to be very long-lasting. This would not really be expected with the putative E2-PKA signaling crosstalk. Can the authors at least offer a speculation for this finding?

(4) Does E2 stimulate the interaction of PKA with BIG3 or only its activity? Fig. 1c gives contradictory results in this regard.

(5) Extended Data Fig. 1h: This result suggests that phosphorylation of BIG3 by PKA is necessary, but not sufficient for the E2- stimulation of PP1. What else is going on? A S305E/S1208E double mutant should be tested.

(6) Stimulation of PKA activity by substrate peptides (lines 137-139): While the data are clear, this statement is confusing. It is not the PKA activity per se that is stimulated, but of course PKA can only phosphorylate a genuine substrate (such as the two serines studied here) and therefore its activity measured.

(7) Impact of PHB2 S39 phosphorylation: according to Extended Data Fig. 2c, the non-phosphorylated form interacts with ER as well, but does not repress. Apart from the fact that Fig. 5 does not illustrate that, this merits a few words of discussion.

Minor comment:

(8) Abstract: The first time S39 is mentioned it is not clear what protein is meant.

(9) Mention that PHB2 is also known as REA.

(10) A scheme of BIG3 (like the one of PHB2 in Fig. 2a) would be helpful.

(11) Indicate in Extended Data Fig. 1f (in the figure itself) which PKA subunit was probed for.

- (12) Fig. 1c: the left panel is apparently mislabelled (should be MCF7?).
- (13) Line 211: Extended Data Fig. 1f should also be referred to for PKA.
- (14) Line 216: "... activation of the PHB2-WT peptide...." ?
- (15) It would be helpful to indicate within the Kaplan-Meier graph of Fig. 4a that this is BIG3 protein and in the corresponding one of Extended Data Fig. 5a that this is BIG3 mRNA (if that's what it is).

Replies to the 1st reviewer's specific comments

1. (The reviewer's comments) A concern is that the PP1C binding property of BIG3 was already identified in a proteomic survey completed by Mattieu Bollen's group (Chem Biol 2009; 16, 365-71). Indeed, the protein is already annotated by the HGNC to be a member of the protein phosphatase regulatory subunit family (designated PPP1R33; <http://www.genenames.org/cgi-bin/genefamilies/set/694>). However, this proteomic survey performed by Bollen's group on the PP1C binding proteins did not interrogate the functional impact of BIG3 as a PPP1R. So, the identification of AKAP and PPP1R properties for BIG3 (along with the influences on downstream signaling) in this study should be considered impactful to the field of ER α -signalling in cancer biology. I would suggest that the authors acknowledge that BIG3 is also annotated as PPP1R33.

(Answer) According to the reviewer's suggestion, we added the following sentences into the Introduction section (Page 6, lines 86–92): Therefore, BIG3 is annotated as *PPP1R33* by the Hugo Gene Nomenclature (HGNC) to be a member of the phosphatase regulatory subunit family. However, the functional impact of BIG3 as a *PPP1R33* remains unknown. Therefore, understanding the properties of BIG3 as an AKAP, including *PPP1R33*, is critical for further elucidating the E2-dependent cell proliferation of ER α -positive breast cancers.

2. (The reviewer's comments) In Fig1B, immunoprecipitations of PP1Ca were completed, and phosphatase activity was measured against pNPP (non-specific substrate). There are no differences in PP1Ca amounts in pull-down, so it is unclear why the authors observe differences in PP1Ca activity. They are actually monitoring total PP1Ca activity against pNPP in the pull-down; the experimental design will not select for phosphatase activity specific to the BIG3-PP1Ca complex. Also there appears to be significant BIG3 pull-down in the absence of any PP1Ca.

Is this immunoreactivity against a non-specific target? The authors should perform the co-IP of BIG3 protein to provide additional substantiation for the presence of an endogenous complex. It would be prudent for the authors to use a substrate that is specific for the BIG3:PP1Ca holophosphatase.

(Answer) According to the reviewer suggestions, to examine phosphatase activity specific to the BIG3-PP1C α complex in breast cancer cells, we first measured the phosphatase activity of PP1C α against pNPP using BIG3 immunoprecipitates, in addition to PP1C α immunoprecipitates in siEGFP-transfected MCF-7 and KPL-3C cells, respectively. Our results showed that co-immunoprecipitation of BIG3 and PP1C α resulted in an E2-dependent increase in PP1C α activity in both BIG3 and PP1C α immunoprecipitates in siEGFP-transfected cells (Fig. 1b and Supplementary Fig. 1d). This description was inserted into the Results section (Page 9, lines 123–126).

Although this reviewer suggested that we should use a substrate that is specific to the BIG3:PP1C α holophosphatase, we have already monitored the free phosphate that was released from the phosphorylated PHB2-S39 peptide as a substrate using an *in vitro* malachite green phosphatase assay. We found that E2 stimulation significantly increased the amount of free phosphate produced in BIG3-immunoprecipitates (Fig. 2e, Supplementary Fig. 3a, and 3c). Moreover, according to the reviewer's suggestion, we performed Phos-tag SDS-PAGE analysis and *in vitro* malachite green phosphatase assays using PHB2 recombinant protein as a substrate, and found that PPC1 α dephosphorylated the phosphorylated PHB2 (Fig. 3d), and remarkably increased the amount of free phosphate produced from phosphorylated PHB2 recombinant protein (Fig. 3e). These descriptions were inserted into the Results section (Page 17, lines 272–277). We appreciate the reviewer's suggestion to improve our manuscript.

3. (The reviewer's comment) In Fig1f, all the IPs from E2 stimulated HEK293T cells have significantly more PP1Ca even though similar amounts of FLAG-BIG3 are present in the pull downs. It is not clear what substrate was employed (presumably pNPP, non-specific phosphatase substrate) or if the authors are normalizing this activity against the PP1Ca signal recovered in the IP. It would be prudent for the authors to use a substrate that is specific for the BIG3:PP1Ca holophosphatase. In Fig 2c-e, why the switch to HEK293T cell system? Is this because the molecular components of the BIG3-PHB2-ER α signaling cascade are not endogenously expressed?

(Answer) As the reviewer commented, in Fig.1f, we measured E2-dependent PPC1 α activity against pNPP as a substrate. Therefore, this phrase was inserted into the Results section (Page 11, line 164) and the figure legend of Fig.1f (Page 53, line 896). In addition, we also inserted the following sentence into the Figure legends section (Pages 53-54, lines 896-898): Each PP1C α activity was normalized to the PP1C α signal intensity of immunoprecipitates in FLAG-tagged BIG3 (WT)-transfected HEK293T cells.

Regarding the measurement of PP1C α activity using a substrate that is specific to BIG3:PP1C α , as described above, we monitored the amount of free phosphate produced from the phosphorylated PHB2 recombinant protein and PHB2-S39 peptide (Fig. 2e, Supplementary Fig. 3a, 3c, and 3e).

Although the reviewer pointed out the reason why we used the HEK293T cell system (Fig.1f and Fig. 2c, d, e), it is very hard to evaluate overexpression experiments using ER α -positive breast cancer cell lines, including MCF-7 and KPL-3C. These cells show high expression of BIG3 and PHB2, but the transfection efficiency of expression vector plasmids into these breast cancer cell lines is quite low (less than 10%). Therefore, because it was impossible to establish stable transformants using these cell lines, we evaluated the effects of PHB2-S39A or BIG3-S305A and S1208A on ERE-luciferase activity, p-IFG1R β phosphorylation

level, and PP1C α activity by co-transfection experiments with several plasmid vectors (ER α , PHB2 or BIG3, etc.) using HEK293T cells with high transfection efficiency in this study.

4. (The reviewer's comment) In Fig. 2g, the authors examine PP1Ca activity toward pS39-PHB2 peptide. It is not surprising that PP1Ca will dephosphorylate this peptide *in vitro*. The experiment does little to offer convincing evidence that a native BIG3-PP1Ca holophosphatase complex preferentially targets the pS39 site of PHB2.

(Answer) According to the reviewer's suggestion, we evaluated PP1C α activity upon the purified full-length recombinant PHB2 phosphorylated at S39 by PKC α using phos-tag SDS-PAGE and *in vitro* malachite green phosphatase assays. The results showed that PPC1 α dephosphorylated the phosphorylated PHB2 (Fig. 3d), and remarkably increased the amount of free phosphate produced from phosphorylated PHB2 (Fig. 3e) as described above. Fig. 2g (Mass-spec data) in the original submitted manuscript was transferred to the revised version as Supplementary Fig. 3e. We have described this result in the Results section (Page 17, lines 272–277).

5. (The reviewer's comment) In Fig. 3b, c, the experimental design lacks rigour. Again, it is not surprising that purified PKCa will phosphorylate S39-PHB2 peptide *in vitro*, so the MS data are not very impactful. The kinase assay employed does not directly assess phosphoryl transfer to the PHB2-peptide substrate. It is unclear from the experiment whether biologically relevant amounts of phosphorylation are observed. To the authors' credit, the follow-up MS experiment does provide validation that the peptide was phosphorylated by PKCa. However, it would be prudent for the authors to provide a true quantitative assessment of PHB2 protein phosphorylation by PKCa (i.e. stoichiometry mol Pi incorporated per mol protein). I

expect the PhosTag-SDS-PAGE technology could be easily adapted to this need [References include: Am J Physiol 2016; 310, C681-C91 and Biochem J 2016; 473;2671-85 and Biochim Biophys Acta 2015; 1854,601-8]. The aa sequence surrounding the S39 residue possesses only weak similarity to the PKC consensus sequence. The KD/KO of PKCa in cell-based experiments could have confounding effects on other kinases (or phosphatases) that influence S39 phosphorylation status.

(Answer) Although the reviewer suggested that it would be prudent for us to provide a true quantitative assessment of PHB2 protein phosphorylation by PKC α through Phos-tag SDS-PAGE technology using gamma-³²P ATP, we agree with his comments. Unfortunately, it is impossible to use radioisotope experiments due to the discarded radiation system in our institute. Hence, on the basis of the reviewer's suggestion, in this study, we examined the direct S39-phosphorylation of recombinant full-length PHB2 protein by PKC α using Phos-tag SDS-PAGE technology and subsequent Western blot analysis with anti-PHB2 and anti-S39-phospho-PHB2 antibodies. Two forms of PHB2, corresponding to unphosphorylated and monophosphorylated PHB2, were clearly detected by Phos-tag SDS-PAGE and Western blot analysis (Fig. 3c). The S39-monophosphorylation of PHB2 by PKC α was saturated at the molar ratio of 1:2 (PHB2: PKC α) (Supplementary Fig. 4d). We have described this result and the Phos-tag SDS-PAGE method in the Results and Methods sections, respectively (Pages 17, lines 265–272; Pages 29–30, lines 486–492). We appreciate the reviewer's suggestion to improve our manuscript.

6. (The reviewer's comment) The use of phosphospecific antibody for pS39-PHB2 is suggestive of its association in nuclear translocation. However, the phosphorylation signal is not quantitative and it is not clear whether a biologically

significant level of PHB2 phosphorylation occurs with ERAP treatments. It is possible that additional or different phosphorylation events may be occurring.

(Answer) We mostly agree with the reviewer's comments. We re-evaluated the effect of S39A–PHB2 on its E2-dependent nuclear–translocation in cancer cells as shown in Fig. 2c. and Supplementary Fig. 2c, d. These results showed that PHB2-WT, but not PHB2-S39A, was released from BIG3 by ERAP treatment to inhibit E2-induced ER α transcriptional activity (Fig. 2c). Immunocytochemical approaches using the anti-PHB2-specific antibody also revealed that phosphorylation of nuclear-translocated endogenous PHB2 at S39 occurred in ERAP-treated MCF-7 cells in the presence of E2, but completely disappeared following λ -phosphatase treatment (Supplementary Fig. 2c). These findings suggest that PHB2-S39 phosphorylation is required for the repression of E2-induced ER α transcriptional activity. On the other hand, we also demonstrated that the amount of PHB2-S39A protein that interacted with nuclear ER α was slightly reduced compared with the amount of PHB2-WT that interacted with nuclear ER α in E2-treated breast cancer cells (Supplementary Fig. 2d), suggesting that PHB2-S39 phosphorylation may also involve the E2-dependent nuclear translocation of PHB2 via possible additional or different phosphorylation events. We have described this evidence in the Results section (Pages 13–14, lines 205–218). Again, we appreciate the reviewer's suggestion to improve our manuscript.

1. (The 1st reviewer's minor comments) Please include the institutional protocol # for human ethics approval.

(Answer) According to the reviewer's comments, we have described the permission number of the human ethical committee of Tokushima University (Permission number #28-16) in the Methods (Page 28, lines 462–464), as well as

the Committee for Safe Handling of Living Modified Organisms in Tokushima University (Permission number 28-5) (Page 26, lines 416–418).

2. (The 1st reviewer's minor comments) There is some difference in N-terminal sequence between KIAA1244 (EAW47918 or AAI37175) and the UniProtKB/Swiss-Prot annotated sequence (Q5TH69). The authors should specify the sequence annotation used for the protein biochemistry.

(Answer) We apologize for the confusion with the sequence annotation of the BIG3 gene. We have described as follows; BIG3 (Q5TH69 in UniProt KB annotation) (Page 5, line 80). In addition, we adopted PP1C α from PP1 α throughout this manuscript.

3. (The 1st reviewer's minor comments) Is PKA detected by WIB – catalytic or regulatory subunit. Since similar amounts of PKA are pulled down with IP of BIG3, why are there distinct levels of pSer and pThr events. Are cAMP levels elevated by E2; hence, greater PKA activity?

(Answer) Thank you very much for the reviewer's comments concerning PKA. We focused on PKA catalytic activity in this study. Therefore, we revised the manuscript concerning the PKA catalytic subunit throughout this manuscript. Moreover, as the reviewer pointed out, the reason why there are distinct levels of pSer and pThr events (Fig. 1d), despite similar amounts of PKA, is that they are pulled down with the IP of BIG3 as shown in Fig. 1c. We re-performed the knockdown of PKA by siRNA because of insufficient knockdown of PKA as shown in Fig. 1d in the original submitted manuscript. As shown in the new Fig. 1d, we demonstrated that depletion of PKA led to almost complete suppression of E2-induced serine and threonine phosphorylation levels of BIG3. Notably, E2-induced PKA kinase activity was drastically enhanced compared with the

E2-induced increased interaction of PKA with BIG3 (Supplementary Fig. 1i, j), suggesting that E2-induced PKA activity is independent of its binding to BIG3. We have described these findings in the Results section (Page 10, lines 148–155).

4. (The 1st reviewer's minor comments) The authors should be careful to distinguish PP1C non-specific activity measures with pNPP vs that measured for the BIG3-PP1C holoenzyme complex.

(Answer) Thank you very much for the reviewer's suggestion. Accordingly, we have distinguished PP1C α non-specific activity measures with pNPP vs that measured for the BIG3-PP1C α holoenzyme complex. We have described pNPP (Fig. 1b, 1e, 1f, Supplementary Fig.1c, 1d, 1e, 1l), phosho-S39 PHB2 peptide (Fig. 2e, Supplementary Fig.3a, 3c), and the phosho-S39 PHB2 recombinant protein (Fig. 3d, 3e, Supplementary Fig.4e) in the Results section.

5. (The 1st reviewer's minor comments) Wording- pg9, line 137-9; how does reaction of S305 and S1208 peptides with recombinant active PKA significantly increase PKA activity?

(Answer) We apologize for this confusion with the description of the data in Fig. 1h. We corrected the sentences as follows: We found that the recombinant PKA phosphorylates the S305 and S1208 peptides, but not the S305A or S1208A peptides (Fig. 1h) (Page 12, lines 176–177).

6. (The 1st reviewer's minor comments) Are other PP1C binding motifs present in BIG3? i.e., G/SILK or MyPhoNe sequences?

(Answer) According to the reviewer's suggestion, we searched other PP1C α binding motifs aside from RVxF in BIG3 by *in silico* analysis; however, there were

no motifs such as G/SILK or MyPhoNe motifs. We have described these findings in the Introduction (Page 6, lines 83–86) and Results (Page 8, line 107–108) sections.

7. (The 1st reviewer's minor comments) Pg10, line 144-146. I am supportive of the authors concluding statement that BIG3 forms a heterotrimeric signaling complex with PP1C and PKA; however, the data that support the “enhancement of PP1C activity in the present of E2 are not as convincing. IP and measurement of BIG3-PP1C phosphatase complex activity against a biologically validated substrate such as PHB2 is required.

(Answer) We agree with the reviewer's comments. We had not yet demonstrated whether the BIG3-PPC1 α -PKA complex enhances PPC1 α activity at that time because we had not used a specific substrate in Figure 1. According to the reviewer's suggestion, we deleted the following sentence “enhancing PPC1 α activity”, and revised as follows: Taken together, these findings strongly suggest that BIG3 forms a heterotrimeric signaling complex with PPC1 α and PKA, and functions as an AKAP in the presence of E2 in breast cancer cells (Page 12, lines 182–184). On the other hand, as shown in Fig. 2e and Supplementary Fig. 3a and 3c, we had already demonstrated that BIG3 immunoprecipitates enhance PP1C α activity in the presence of E2 using phospho-PHB2 peptide as a substrate. We appreciate the reviewer's suggestion to improve our manuscript.

8. (The 1st reviewer's minor comments) P11, line 160; spelling “phosphoarylated”

(Answer) We apologize for the spelling error. We corrected the spelling (Page 13, line 198). We appreciate the reviewer's suggestion to improve our manuscript.

9. (The 1st reviewer's minor comments) P12, line 192-3; “indicating BIG3

phosphorylation enhanced phosphatase activity” Specifically it appears from the data in Fig 2e that both S305 and S1208 phosphorylations are required to achieve maximal PHB2 phosphatase activity.

(Answer) According to the reviewer’s suggestion, we revised the sentences as follows: “...indicating that both S305 and S1208 phosphorylation are required to achieve maximal PHB2 phosphatase activity” (Pages 15, lines 234–236). We appreciate the reviewer’s suggestion to improve our manuscript.

10. (The 1st reviewer’s minor comments) Wording-Pg14, line 216; “the significant activation of the PHB2-WT peptide”. What is the measure of peptide “activity”? Do the authors just mean that PKC can phosphorylate the PHB2 peptide?

(Answer) We apologize for this confusion with the description of the data in Fig. 3b. We corrected the sentences as follows: Subsequent *in vitro* kinase assays showed that PKC α phosphorylated the PHB2-WT peptide, but only minimally phosphorylated the PHB2-S39A peptide (Fig. 3b) (Page 16, lines 259–261).

11. (The 1st reviewer’s minor comments) Were investigators blinded to classification of IHC pathology slides?

(Answer) Yes, we had already described this in the Methods section as follows: Experienced pathologists without prior knowledge of clinicopathological data semi-quantitatively evaluated the intensity of BIG3 and PHB2 phosphorylation staining based on the following scoring system: 0, negative (no staining); 1, weak; 2, moderate; and 3, positive (Page 35, lines 577–582).

12. (The 1st reviewer’s minor comments) Pg16, line 257; “a novel cancer-specific AKAP” BIG3 should also be classified as a PPP1R targeting subunit unless the

authors feel the dominant influence of BIG3 is on the kinase signaling.

(Answer) According to the reviewer's suggestion, the following sentences were inserted into the Discussion section (Page 20, lines 329–332). In the present study, we demonstrated for the first time that BIG3 forms a tri-complex with upstream kinase PKA and a regulatory subunit of PPC1 α to function as a novel cancer-specific AKAP, which is classified as a PPP1R targeting subunit. We appreciate the reviewer's suggestion to improve our manuscript.

13. (The 1st reviewer's minor comments) Is the targeting function of BIG3 for PP1C and PKA also conserved among the other ARFGEF members (BIG1 and BIG2) or is this a unique architectural feature of the BIG3 protein? Interestingly, the KAVSF (KVxF) motif that provides PP1C-binding appears to be absent in the plant variant of BIG3 (*A. thaliana*: Q9LPC5.1: although a KVLFD sequence is present at residue 1355).

(Answer) Thank you for the reviewer's comments. Indeed, BIG3 contains a canonical PPC1 α binding motif 'RVxF' sequence¹⁵ as well as BIG1 and BIG2. On the other hand, *in silico* analysis showed that BIG3 has no other PP1C binding motifs such as G/SILK and MyPhoNe, but both BIG1 and BIG2 contain the G/SILK motif. These descriptions were inserted into the Introduction section (Page 6, lines 83–86).

14. (The 1st reviewer's minor comments) Pg19, line313; "does not have an inhibitory effect" Do the authors mean a disruptive effect on holophosphatase complex stability?

(Answer) We apologize for the confusion with this description. We would like to delete this description because this is an incorrect interpretation. We revised the

following sentences: "...indicating that E2-induced PKA-mediated phosphorylation of BIG3-S305 and -S1208 may result in a conformational change, exposing an inhibitory surface of PPC1 α , although further structural studies are required to elucidate the precise mechanism by which BIG3 inhibits PPC1 α activity." (Page 24, lines 393–396).

15. (The 1st reviewer's minor comments) Fig2e; pS305-BIG3 mislabelled as "pS1305"

(Answer) We apologize for the error of mislabeling pS305-BIG3. We corrected the Western blot panel in Fig. 2e. We appreciate the reviewer's suggestion to improve our manuscript.

16. (The 1st reviewer's minor comments) Fig3a; Were WCL actually used for loading controls or do the WCLs represent total cytoplasm and nuclear fractions used for the IPs.

(Answer) According to the reviewer's comments, we inserted "C-WCL and N-WCL, which show total cytoplasmic lysates and total nuclear lysates, respectively) into Fig. 3a and the figure legend of Fig. 3a (Page 56, lines 933–934). We appreciate the reviewer's suggestion to improve our manuscript.

Replies to the 2nd reviewer's specific comments

1. (The 2nd reviewer's major comments) Connection with ER (see also comment #7): while some consequences for ER activity have been looked at, it is not very clear how the known ER-PHB2 interaction fits into this story. That ERAP promotes the phosphorylation of PHB2 and thereby repression is well documented here, but this is obviously a pharmacological situation. Could it be that ER is also a component of the tri-complex? Perhaps through PKA or even PP1? What's confusing is that the

hormone estrogen (E2) both promotes the inactivation of the repressor PHB2 (through PKA) and activates it (through PKC). What happens in the real world (without ERAP) when both pathways are turned on simultaneously by E2? Is it a question of balance? Perhaps playing with that by overexpressing or knocking down the two kinases downstream of E2 might clarify this.

(Answer) Thank you very much for the reviewer's comments. This is a very critical point in this study. According to the reviewer's suggestions, to solve the conflict that E2 treatment caused the inactivation of PHB2 via BIG3-PKA-PP1 α , but led to the activation of PHB2 via PKC α , we examined the impact of PKA and PKC α knockdown on the S39-phosphorylation level of PHB2 and the interaction of BIG3 with PKC α , PKA, PHB2, and ER α in the presence of E2 in MCF-7 cells. The results showed that PKA depletion resulted in the recovery of E2-induced S39-phosphorylation of PHB2 that was immunoprecipitated with BIG3 and the weak interaction of BIG3 with PKC α for 12 h after E2 treatment. On the other hand, PKC α depletion resulted in no effect on the S39-phosphorylation of PHB2 or the BIG3-PKA and BIG3-PHB2 interaction (Fig. 3f, Supplementary Fig. 4f). Notably, we observed no interaction of BIG3 with ER α via PHB2 despite E2 stimulation in PKA- or PKC α -depleted MCF-7 cells, as well as siEGFP-treated cells (Fig. 3f). Therefore, BIG3-PKA-PP1C α tri-complexes preferentially sustain PHB2 in a dephosphorylated, inactive state even for E2 stimulation, suggesting the possibility that BIG3-PKA-PP1C α tri-complexes inhibit the approach of ER α and PKC α . We have described these findings in the Results section (Pages 18, lines 282–297).

Taken together, we propose that, in ER α -positive breast cancer cells, E2 stimulation induces PKA-dependent S305 and S1208 phosphorylation in BIG3, which cancels its negative regulation of PPC1 α activity, resulting in the apparent inactivation of PHB2 through S39 dephosphorylation via the interference of the interaction of PKC α with the BIG3-PKA-PPC1 α complex. This is likely due to conformation changes in the PKC α -binding region(s) of PHB2 (Fig. 5, upper panel).

We have described these sentences in the Results section (Pages 20-21, lines 332–337).

2. (The 2nd reviewer's major comments) Several IPs lack negative controls: Fig. 1a, Fig. 2d, Extended Data Fig. 1b, Extended Data Fig. 2c.

(Answer) We apologize for the lack of negative control (IgG) in several IP experiments. We re-performed IP experiments using IgG as a negative control, and inserted the results into Fig. 1a, Fig. 2d, Supplementary Fig. 1b, and Supplementary Fig. 2d in the revised manuscript.

3. (The 2nd reviewer's major comments) E2 activation of PP1 seems to be very long-lasting. This would not really be expected with the putative E2-PKA signaling crosstalk. Can the authors at least offer a speculation for this finding?

(Answer) According to the reviewer's suggestion, to clarify the effect of E2-PKA signaling on PP1C α (former name PP1 α) activity, we examined BIG3-S305 and -S1208 phosphorylation levels at 12 and 24 hours after E2 stimulation. The results showed that the E2-induced phosphorylation level of BIG3-S305 and -S1208 was maintained for 24 h in MCF-7 cells (Fig. 1g), whereas it was almost completely suppressed by siPKA treatment (Supplementary Fig. 1m). As the reviewer commented, we demonstrated that E2-induced increased PPC1 α activity that was maintained for 24 h in MCF-7 and KPL-3C cells (Supplementary Fig. 1e). According to this evidence, putative E2-PKA signaling can activate PPC1 α activation at least for 24 h in breast cancer cells. We have described these results in the Results section (Page 11, lines 168–173).

More importantly, we noticed that PP1C α expression was remarkably upregulated after E2 stimulation, whereas it was inhibited by treatment with

tamoxifen, a selective ER α modulator, at both the protein and transcriptional levels as observed from Western blotting (Supplementary Fig. 1f). To obtain direct evidence for the upregulation of *PPP1CA* (gene name of PP1C α) expression by E2 treatment, we performed a ChIP assay with E2-stimulated MCF-7 cells. The results showed that E2-dependent ER α recruitment was associated with the 5'-oestrogen response element (ERE) of the *PPP1CA* gene in MCF-7 cells (Supplementary Fig. 1f). As expected, E2 stimulation resulted in robust luciferase activity in cells transfected with the construct containing the 5'-ERE from *PPP1CA* (Supplementary Fig. 1f), suggesting the possibility that PP1C α activation may be very long-lasting because it is a potential ER α -target gene. We would like to add these new findings in the Results section (Pages 9–10, lines 130–141).

4. (The 2nd reviewer's major comments) Does E2 stimulate the interaction of PKA with BIG3 or only its activity? Fig. 1c gives contradictory results in this regard.

(Answer) The reviewer pointed out whether E2 stimulation leads to increased PKA-BIG3 interaction or enhancement of PKA activity. To answer this question, we compared the amount of BIG3 binding to PKA with PKA activity after E2 stimulation in breast cancer cells. The results showed that E2-induced PKA kinase activity was drastically enhanced compared with the E2-induced increased interaction of PKA with BIG3 (Supplementary Fig. 1i, j), suggesting that E2 stimulation led to the enhancement of PKA activity rather than the increase of its binding to BIG3. We have described these findings in the Results section (Page 10, lines 151–155).

5. (The 2nd reviewer's major comments) Extended Data Fig. 1h: This result suggests that phosphorylation of BIG3 by PKA is necessary, but not sufficient for the E2-

stimulation of PP1. What else is going on? A S305E/S1208E double mutant should be tested.

(Answer) According to the reviewer's suggestion, we made a S305E/S1208E double mutant construct, and then re-evaluated the effect of the S305E/S1208E double mutant, as well as other mutants (S305A, S305E, S1208A and S1208E), on PP1C α activity against pNPP as a substrate. Among these sites, mutations in both S305 and S1208 (via substitutions with alanine) significantly reduced E2-dependent PPC1 α activity against pNPP as a substrate (Fig. 1f and Supplementary Fig. 1l). In contrast, pseudo-phosphorylation mutations at each serine residue (S305E and S1208E), and, in particular, double pseudo-phosphorylation mutations (S305E/S1208E) did not completely reduce PPC1 α activity (Supplementary Fig. 1l), suggesting that S305 and S1208 phosphorylation of BIG3 by PKA may be necessary for E2-dependent PPC1 α activity. We have described these findings in the Results section (Page 11, lines 162–168).

6. (The 2nd reviewer's major comments) Stimulation of PKA activity by substrate peptides (lines 137-139): While the data are clear, this statement is confusing. It is not the PKA activity per se that is stimulated, but of course PKA can only phosphorylate a genuine substrate (such as the two serines studied here) and therefore its activity measured.

(Answer) We apologize for the confusion regarding the statement of the data in Fig. 1h. We corrected the description as follows: We found that recombinant PKA phosphorylates the S305 and S1208 peptides, but not the S305A or S1208A peptides (Fig. 1h) (Pages 12, lines 176–177).

7. (The 2nd reviewer's major comments) Impact of PHB2 S39 phosphorylation: according to Extended Data Fig. 2c, the non-phosphorylated form interacts with ER as well, but does not repress. Apart from the fact that Fig. 5 does not illustrate that, this merits a few words of discussion.

(Answer) We re-evaluated the effect of S39A-PHB2 on its E2-dependent nuclear-translocation and its E2-dependent repression activity in cancer cells as shown in Fig. 2c and Supplementary Fig. 2c, d. These results showed that PHB2-WT, but not PHB2-S39A, released from BIG3 by ERAP treatment inhibited E2-induced ER α transcriptional activity (Fig. 2c). Immunocytochemical approaches using the anti-PHB2-specific antibody also revealed that phosphorylation of nuclear-translocated endogenous PHB2 at S39 occurred in ERAP-treated MCF-7 cells in the presence of E2, but completely disappeared following λ -phosphatase treatment (Supplementary Fig. 2c; Extended Data Fig. 2d in original submitted version). These findings suggest that PHB2-S39 phosphorylation is required for the repression of E2-induced ER α transcriptional activity. On the other hand, we also demonstrated that the amount of PHB2-S39A protein that interacted with nuclear ER α was slightly reduced compared with the amount of PHB2-WT that interacted with nuclear ER α in E2-treated breast cancer cells (Supplementary Fig. 2d; Extended Data Fig. 2c in original submitted version). These findings suggest that PHB2-S39 phosphorylation may also involve the E2-dependent nuclear translocation of PHB2 via possible additional or different phosphorylation events. We have described this evidence in the Results section (Pages 13–14, lines 205–218). We appreciate the reviewer's suggestion to improve our manuscript. Furthermore, according to the reviewer's suggestion, we inserted the following sentence into the Discussion section; Of note, the non-phosphorylated form of PHB2 (such as S39A) interacts with ER α as well, but does not repress signaling pathways (Page 21, lines 341–344). We appreciate the reviewer's suggestion to improve our manuscript.

8. (The 2nd reviewer's minor comments) Abstract: The first time S39 is mentioned it is not clear what protein is meant.

(Answer) According to the reviewer's suggestion, we inserted "of PHB2" into the Abstract section (Page 3, lines 38–39). We appreciate the reviewer's suggestion to improve our manuscript.

9. (The 2nd reviewer's minor comments) Mention that PHB2 is also known as REA.

(Answer) According to the reviewer's suggestion, we inserted "which is also known as REA" into the Introduction section (Page 4, line 56). We appreciate the reviewer's suggestion to improve our manuscript.

10. (The 2nd reviewer's minor comments) A scheme of BIG3 (like the one of PHB2 in Fig. 2a) would be helpful.

(Answer) According to the reviewer's suggestion, we inserted the scheme of the BIG3 protein as a new Fig. 1i in the Results section (Page 12, lines 182–184). We appreciate the reviewer's suggestion to improve our manuscript.

11. (The 2nd reviewer's minor comments) Indicate in Extended Data Fig. 1f (in the figure itself) which PKA subunit was probed for.

(Answer) Thank you very much for your comments concerning PKA. We focused on PKA catalytic activity in this study. Therefore, we revised the text regarding the PKA catalytic subunit (from PKA) in all of the immune blot data (Fig. 1c, 1d, 1f, 2e, 2f, and Supplementary Fig. 1m, 3a, 3c, 4b). We appreciate the reviewer's suggestion to improve our manuscript.

12. (The 2nd reviewer's minor comments) Fig. 1c: the left panel is apparently mislabelled (should be MCF7?).

(Answer) We apologize for the error mislabeling the cell line name. We described "MCF-7" in the right panel of Fig. 1c (Page 10, line 146). We appreciate the reviewer's suggestion to improve our manuscript.

13. (The 2nd reviewer's minor comments) Line 211: Extended Data Fig. 1f should also be referred to for PKA.

(Answer) We apologize for no indication of the Extended Data Fig. 1f (original submitted version) for PKA expression in the cell lines. We inserted Supplementary Fig. 1h (revised manuscript) into Page 16, line 254. We appreciate the reviewer's suggestion to improve our manuscript.

14. (The 2nd reviewer's minor comments) Line 216: "... activation of the PHB2-WT peptide...." ?

(Answer) We apologize for this confusion regarding the description of the data in Fig. 3b. We corrected the sentences as follows: Subsequent *in vitro* kinase assays showed that PKC α phosphorylated the PHB2-WT peptide, but only minimally phosphorylated the PHB2-S39A peptide (Fig. 3b) (Page 16, lines 259–261).

15. (The 2nd reviewer's minor comments) It would be helpful to indicate within the Kaplan-Meier graph of Fig. 4a that this is BIG3 protein and in the corresponding one of Extended Data Fig. 5a that this is BIG3 mRNA (if that's what it is).

(Answer) According to the reviewer's suggestion, we described the "BIG3 protein" in the Kaplan-Meier graph of Fig. 4a and Supplementary Fig. 5a, and also the "BIG3 mRNA" in the Kaplan-Meier graph of Supplementary Fig. 5a. We appreciate the reviewer's suggestion to improve our manuscript.

REVIEWERS' COMMENTS:

Reviewer #1 (Remarks to the Author):

The authors have responded constructively to my original concerns. Additional experimental results and/or appropriate clarifications to written materials were provided.

1. The provision of new experimental results with Phos-Tag gels demonstrating the substantive impact of BIG3:PP1Ca effect on PHB2-pS39 phosphorylation status help to strengthen the authors conclusions.

2. The presence of more PP1Ca in pulldowns of Flag-BIG3 from E2-stimulated HEK293T cells is still puzzling (Figure 1f). The authors have dealt with the situation by normalizing the measured activity toward pNPP against the amount of PP1Ca in the pull-down. This may elevate the specific activity calculated for the E2(-) samples, but I do not think it would impact upon the significant finding that BIG3 phosphorylation is associated with a reduction in activity. There is also substantively more PP1Ca in the pull-downs from E2(+) samples for the S305A and S1208A mutants.

All minor concerns were addressed in the revised manuscript; however, the authors may wish to revise comments on the possible mechanisms for the effect of BIG3 phosphorylation on the holoenzyme activity. (as a follow-up to comment #14 regarding the statement made on pg 19, line 313).

The exposure of an inhibitory surface of PP1Ca by phosphorylation of BIG3-S305 and S1208 is possible; however, I am unaware of such a conformational change in the PP1C catalytic subunit having been reported in the literature with phosphorylation of a PP1-regulatory/targeting protein. Inhibitory phosphorylations associated with other PP1-regulatory/targeting holoenzymes are known to involve auto-inhibition (e.g. the production of a potential autoinhibitory site - mimicking the phospho-substrate). Alternatively, phosphorylation of PP1-regulatory/targeting proteins may release the phosphatase holoenzyme from the intracellular region associated with substrate availability. As the authors' rightly suggest - additional structural/functional studies will be required to identify the mechanism whereby BIG3 phosphorylation inhibits holoenzyme activity.

Sincerely
Justin MacDonald (University of Calgary)

Reviewer #2 (Remarks to the Author):

The manuscript has been extensively revised. It includes a lot of new data and the text has been edited. My comments have been satisfactorily addressed.

Replies to the 1st reviewer's specific comments

1. (The reviewer's comments) The provision of new experimental results with Phos-Tag gels demonstrating the substantive impact of BIG3:PP1Ca effect on PHB2-pS39 phosphorylation status help to strengthen the authors conclusions.

(Answer) According to the reviewer's suggestion, we added the following sentence into the conclusion of Discussion section (Page 25, lines 420–421): BIG3-PKA-PP1C α tri-complexes preferentially inactivate PHB2 via PHB2-S39 dephosphorylation in the presence of E2 stimulation. We appreciate the reviewer's suggestion to improve our manuscript.

2. (The reviewer's comments) The presence of more PP1Ca in pulldowns of Flag-BIG3 from E2-stimulated HEK293T cells is still puzzling (Figure 1f). The authors have dealt with the situation by normalizing the measured activity toward pNPP against the amount of PP1Ca in the pull-down. This may elevate the specific activity calculated for the E2(-) samples, but I do not think it would impact upon the significant finding that BIG3 phosphorylation is associated with a reduction in activity. There is also substantively more PP1Ca in the pull-downs from E2(+) samples for the S305A and S1208A mutants.

(Answer) We apologize for his confusion against the interpretation of data in Figure 1f. We agree with the reviewer's comments concerning Figure 1f. We confirmed that the amounts of PP1C α which bound to exogenously expressed FLAG-BIG3 in HEK293T cells was obviously increased in the presence of E2 stimulation. This result suggests the possibility that the E2-dependent PP1C α phosphatase activation was depended on the increase of PP1C α amounts which binds to BIG3. On the other hand, mutations in both S305 and S1208 of BIG3, but not mutations in other sites significantly reduced E2-dependent PPC1 α activity regardless of high amount of PP1C which binds to BIG3. These findings suggest that both S305 and S1208 of BIG3 are important sites for PP1C α phosphatase activity. The above description was inserted into the Results section (Page 11, lines 162–172). We appreciate the reviewer's suggestion to improve our manuscript.

In fact, the increase of PP1C α amounts which were pull-downed with Flag-BIG3 under the E2 stimulation may due to exogenous experiment systems because there is no difference of amount of endogenous PP1C α which bound to BIG3 with or without E2 stimulation as shown in Fig. 1b. To overcome this issue, we should measure PP1C α phosphatase activity using the wildtype, S305A and S1208A of BIG3 full-length recombinant proteins. However, it is very hard at present to prepare the full-length recombinant BIG3 proteins because BIG3 protein is huge protein (2177 amino acids). Therefore, we would like to attempt to prepare the recombinant BIG3 proteins in next issue.

3. (The reviewer's comments) All minor concerns were addressed in the revised manuscript; however, the authors may wish to revise comments on the possible mechanisms for the effect of BIG3 phosphorylation on the holoenzyme activity. (as a follow-up to comment #14 regarding the statement made on pg 19, line 313).

The exposure of an inhibitory surface of PP1Ca by phosphorylation of BIG3-S305 and S1208 is possible; however, I am unaware of such a conformational change in the PP1C catalytic subunit having been reported in the literature with phosphorylation of a PP1-regulatory/targeting protein. Inhibitory phosphorylations associated with other PP1-regulatory/targeting holoenzymes are known to involve auto-inhibition (e.g. the production of a potential autoinhibitory site - mimicing the phospho-substrate). Alternatively, phosphorylation of PP1-regulatory/targeting proteins may release the phosphatase holoenzyme from the intracellular region associated with substrate availability. As the authors' rightly suggest - additional structural/functional studies will be required to identify the mechanism whereby BIG3 phosphorylation inhibits holoenzyme activity.

(Answer) According to the reviewer's suggestion, we inserted the description about the possibility of auto-inhibition in response to the phosphorylations of holoenzymes themselves into the Discussions section (Page 24, lines 404–407) by adding the relating references. We appreciate the reviewer's suggestion to improve our manuscript.